# Relating natural image statistics to patterns of response covariability in macaque primary visual cortex

Amirhossein Farzmahdi [1,4] ✉, Adam Kohn[1,2,3] & Ruben Coen-Cagli [1,2,3] ✉

Determining how the brain encodes sensory information requires understanding the structure of cortical activity, including how its variability is shared among neurons. The role of this covariability in cortical representations of natural visual inputs is unclear. Here, we adopt the neural sampling hypothesis and extend a well-established generative model of image statistics, to explain pairwise activity as representing joint probabilistic inferences about latent features of images. According to the theory, variability reflects uncertainty about those latent features. In natural images, some sources of uncertainty are shared between features and lead to covariability between neurons, whereas other independent sources contribute to private variability. Our analysis shows that spatial context in images reduces shared uncertainty for overlapping features, whereas it reduces independent uncertainty for nonoverlapping features. As a result, the model predicts that increasing the size of an image reduces correlations for pairs with overlapping receptive fields and increases correlations for pairs with offset receptive fields. This prediction was confirmed by recordings from male macaque primary visual cortex (V1). Our study establishes a precise connection between V1 correlations and natural scene statistics, suggesting patterns of covariability are a feature of probabilistic representations of scenes.

Understanding how visual cortical neurons represent natural stimuli is a major goal in neuroscience. Progress in this field has been supported by normative theories that predict how neurons ought to encode visual stimuli to achieve computational objectives such as coding efficiency, probabilistic inference, or object recognition[1–3], and by related data-analytic tools[4].

While traditional approaches have often focused on explaining single-neuron mean firing rates, there is a growing recognition that the cortical neural code for images is distributed across large populations. Therefore, understanding the encoding of scenes requires understanding the interactions between neurons[5,6]. Whether theories developed for single-neuron responses to natural images generalize to the structure of neural population activity remains largely unexplored.

Prominent studies of the neural encoding of parametric simple visual stimuli have demonstrated the importance of neural interactions, placing much emphasis on trial-by-trial variability shared between neurons, i.e., *correlations* between the activity fluctuations of pairs of neurons responding to a fixed stimulus (often termed noise correlations, spike-count correlations, or $r_{sc}$[7]). This is because correlated variability can determine the information encoded by a neural population about parametric stimuli[8–19]. However, extending this framework to complex natural inputs encompassing multiple features is challenging[20].

[1]Department of Systems and Computational Biology, Albert Einstein College of Medicine, Bronx, NY, USA. [2]Dominick P. Purpura Department of Neuroscience, Albert Einstein College of Medicine, Bronx, NY, USA. [3]Department of Ophthalmology and Visual Sciences, Albert Einstein College of Medicine, Bronx, NY, USA. [4]Present address: Zuckerman Institute, Columbia University, New York, NY, USA. ✉e-mail: af3587@columbia.edu; ruben.coen-cagli@einsteinmed.edu

To address these problems, here we extend a well-established theory of V1 encoding, to generate new predictions for V1 covariability in response to natural stimuli and test them with recordings from macaque V1. The theory posits that the goal of V1 neurons is to represent probabilistic inferences about low-level features of images[21]. Testing the theory requires specifying a generative model of the statistics of those features in natural images, and, given a visual input, inverting the generative model to compute a posterior distribution over the latent features. In line with much prior work[22–27] (see ref. 28 for review), here we consider a simple generative model known as Gaussian Scale Mixture (GSM[29]). The key assumption of this model is that a global 'modulator' variable modulates multiple features and thus introduces statistical dependence among them (details in Fig. 1A and in Results). The second element of the theory is an assumption about how neural activity represents the inferences. We adopt the sampling hypothesis, according to which instantaneous activity of a neuron represents a sample from the posterior distribution[20,25]. It follows that the across-trial mean and variance of the activity of a neuron, given an input stimulus, reflect the mean and variance (uncertainty) of the posterior distribution (Fig. 1C).

Past work strongly supports this theoretical framework—which combines a GSM model of natural image statistics with the sampling hypothesis of neural representation—for explaining single neuron activity[2,23,25–27,30,31] including that driven by natural images[2,27], and there is evidence that the theory may reproduce properties of V1 covariability[25,26,32–35]. Building on this foundation, we focus on how response covariability is modulated by image manipulations for which the single-neuron theory has made predictions that were confirmed experimentally: nonlinear contextual modulation, or modulation of the response to a target stimulus by presentation of a surrounding stimulus[36–40]. We hypothesize that inferences about pairs of features should be modulated by spatial context in images, because contextual stimuli reveal new information about a stimulus and therefore reduce uncertainty. Importantly, our analysis of natural image statistics indicates multiple sources of uncertainty: Some sources are shared between features (Fig. 1B, top) and thus induce correlated variability, whereas others are independent (Fig. 1B, bottom) and thus induce independent variability. We further show that similar and overlapping features tend to have shared uncertainty, hence pairwise correlations between neurons encoding those features are reduced as the image is made larger by adding spatial context. Conversely, larger images reduce independent variability, thus increasing correlations, between neurons with less overlapping features. This prediction is strongly supported by our analysis of macaque V1 responses to natural images.

## Results

### Pairwise models with shared versus independent latent modulators to study V1 correlations

To study the relationship between pairwise neural responses and image statistics under the theory of probabilistic inference, we implemented pairwise Gaussian Scale Mixtures (GSM) generative models of image statistics. We then inverted the generative model to infer the posterior distributions of latent variables. Lastly, to establish a link between the inferred distribution and neural activity, we adopted the neural sampling hypothesis (Fig. 1).

The GSM model for a single neuron (Fig. 1A) assumes that an image is generated from linear combinations of localized oriented features (each feature is like an elementary image: a wavelet with a specific orientation and spatial frequency), each weighted by a Gaussian coefficient (a latent variable $g$). A global modulator variable (latent variable $v$) scales multiplicatively that weighted sum, and noise ($\eta$) is added, resulting in the observed variable $x$ (related to the image by a linear transformation; see Methods). This is a mathematical description of how an image may be generated: each choice of specific values for the coefficients, the modulator, and the noise, will generate one specific image.

The problem faced by V1 neurons, we hypothesize, is the inverse problem: when an image is presented (the visual input), we assume that V1 neurons infer the values of the coefficients that are likely to have generated that particular image. More precisely, we hypothesize that a V1 neuron encodes the posterior distribution of the Gaussian

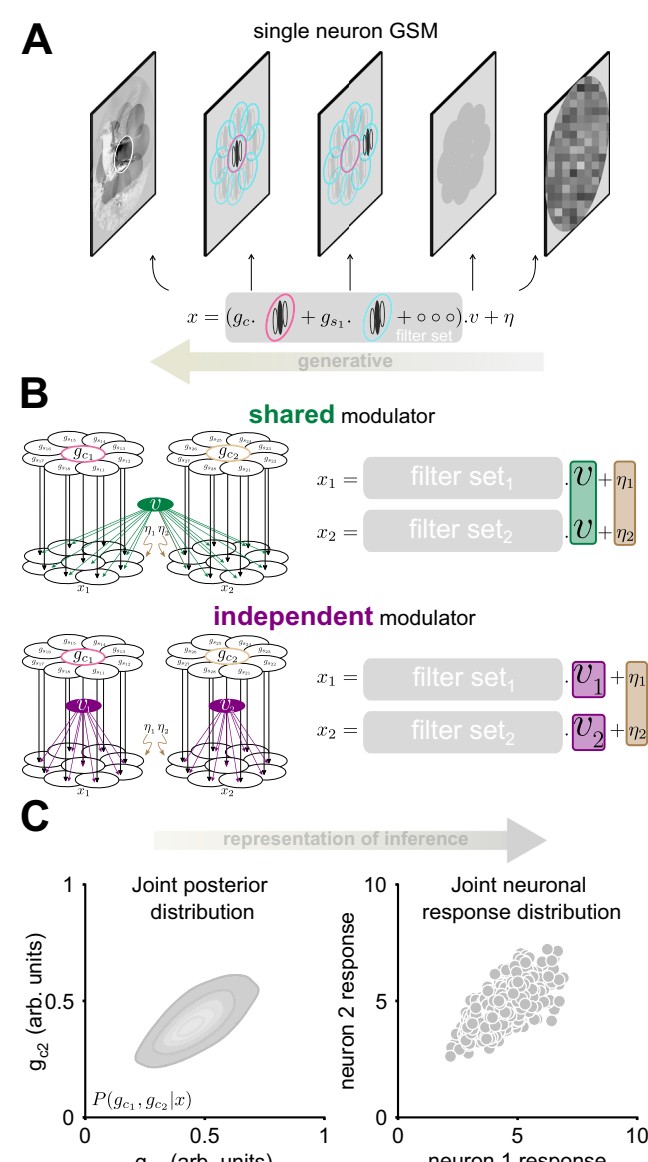

**Fig. 1 | Overview of the theory: pairwise GSM models for neural covariability. A** A summary of the generative process of the Gaussian scale mixture (GSM) model. The linear transform of the raw image pixel values, denoted as $x$, results from combining local oriented features (in pink and cyan), each weighted by a Gaussian coefficient $g$. These weighted features (denoted `filter set') are then collectively scaled by a global modulator $v$ and the result is corrupted by additive Gaussian noise, denoted as $\eta$. **B** The schematic illustrates the generative process of the pairwise shared and independent GSM models. Each distinct filter set refers to one neuron. In the top row, a single global modulator, $v$, is shared between the two model neurons (the green patch in the right equation), defining the shared modulator model (shared GSM). Conversely, in the bottom row, the independent modulator model (independent GSM) is characterized by individual global modulators, $v_1$ and $v_2$, highlighted by two distinct purple color patches for each neuron. **C** In our theory, a pair of model neurons encodes the joint posterior distributions of their features (left). According to the neural sampling hypothesis, neural activity corresponds to samples from this joint posterior distribution. Multiple samples correspond to independent measurements over time or stimulus repetitions (right).

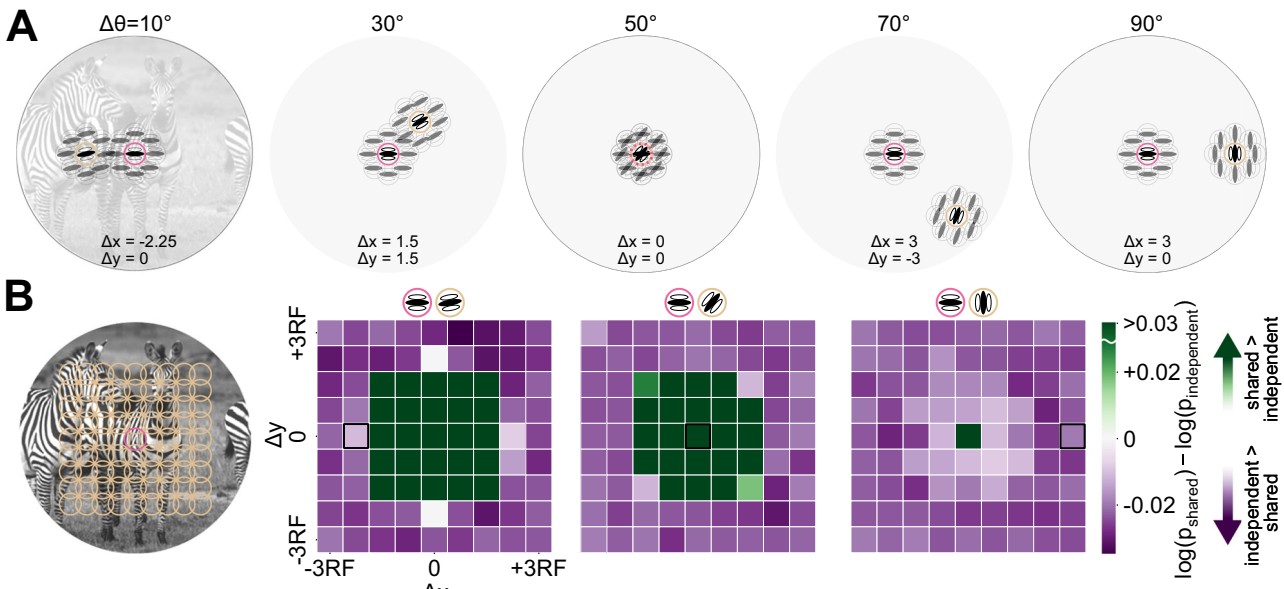

**Fig. 2 | Optimal pairwise GSM structure for natural images depends on receptive fields distance and dissimilarity. A** Schematic representation of five example pairs of model neurons. Each neuron is represented by a set of filters covering the receptive field center and surround. When computing model responses to a natural image, one reference neuron is kept at the central location and horizontal orientation (pink), while the orientation and location of the other are changed (beige). $\Delta\theta$ denotes the difference in orientation preferences, which are indicated above each one of the example pairs (ranging from 10° to 90°). **B** Leftmost panel: we considered neural pairs with one reference neuron (pink circle) and the second neuron centered at each position on the grid of yellow circles (horizontal and vertical displacements from the reference neuron ranging from −3 to 3 times the RF). Right three panels: The graphs compare the log-likelihood of natural images under the shared versus independent GSM models. Log-likelihood ratios, mean over testing images, are shown for each pair of locations and for three example $\Delta\theta$ values (reference at 90° and second neuron at 80, 50, and 0° from left to right). The log-likelihood ratio quantifies which pairwise GSM model better captures the statistics of natural images (positive values, in green, indicate that the shared modulator is better, and negative in purple, that the independent modulator is better). For each pair of neurons, the GSM parameters (i.e., covariance matrices) were optimized using 10,000 natural and 10,000 white noise training images, and the likelihood was estimated on a non-overlapping set of 10,000 test images (see Methods). The three locations outlined in black correspond to the example pairs outlined in panel (**A**). Positive values (shared modulator better) are cropped to the same range as negative values for better visualization. When the receptive fields overlap, the ratio is distinctly positive, indicating that the shared modulator model captures image statistics better. In contrast, with non-overlapping receptive fields, the ratio turns mostly negative, particularly for larger values of $\Delta\theta$, suggesting that the independent GSM model more effectively captures image statistics.

coefficient ($g$) associated with a target feature (Eqs. (1) and (2) in Methods).

Given our focus on pairwise V1 activity, our primary objective is to estimate the joint posterior probability distribution of the features encoded by two neurons, given an image, denoted as $p(g_{c1}, g_{c2}|x)$. The numbers indicate the model neuron, and $c$ denotes centered features ($g_{c1}$: pink, $g_{c2}$: beige in Fig. 1B, left). The joint responses of neuron pairs to the same image are interpreted as samples from such a distribution (Fig. 1C). To construct this distribution, we therefore considered pairs of model neurons similar to the above, except that we allowed for two distinct structures. In the first structure, termed the *shared pairwise GSM*, the global modulator is shared between all the coefficients of both model neurons ($v$; Fig. 1B, top). In the second structure, termed the *independent pairwise GSM*, two independent modulators are used, one for each neuron ($v_1, v_2$; Fig. 1B, bottom).

The use of these two distinct GSMs was motivated by observations on the statistical properties of natural images, which often exhibit non-stationarities: namely, statistical dependencies can vary across regions. Due to these non-stationarities, features within the same visual object tend to be statistically dependent, influenced by common underlying factors. These dependencies are effectively captured by a GSM model with a single global modulator that scales all features of an object. In contrast, features belonging to different visual objects are generally more independent, as they are influenced by separate factors[31]. For such cases, using independent modulators that scale each feature separately, better captures the statistical independence of those features.

For the pairwise application considered here, we reasoned that joint inferences about pairs of features (represented by pairs of neurons) should account for whether these features are a priori more likely to be part of the same or different visual objects. This prior probability depends on factors such as similarity (e.g., orientation preference) and spatial proximity. Intuitively, features that are similar and located close together are more likely to belong to the same object and are thus better modeled with a shared global modulator. Conversely, dissimilar or distant features are more likely to belong to different objects, making independent modulators more suitable. By incorporating this reasoning into our model, we aim to capture the statistical dependencies (or lack thereof) between different features in natural images. Next, we tested this intuition formally.

## Image statistics are captured by shared or independent modulators, depending on the proximity and similarity of image features

To study how well the shared and independent GSM capture image statistics, we computed the likelihood of each natural image under each model, and compared the models by their log-likelihood ratio. We implemented the GSM for each neuron similarly to past work on surround modulation, i.e., with a group of bandpass linear filters covering a reference location and eight surrounding locations (Fig. 2A; details in Methods). The model parameters, i.e., the prior covariance matrices of the local latent features, were estimated with moment matching[41] from an ensemble of 10,000 natural images from the ImageNet validation set[42]. This moment-matched covariance matrix is practically equivalent to the maximum likelihood estimate (MLE), and thus the resulting marginal likelihoods in Fig. 2 can be interpreted as approximating maximum marginal likelihoods. The log-likelihood was measured on a

separate, randomly selected subset of 10,000 natural images from the ImageNet test set. For a given test image, the likelihood value of each model indicates the effectiveness of the models in capturing the image statistics (computed as in ref. 31). To investigate the covariability between two model neurons, we systematically varied their tuning similarity (relative orientation preference) and proximity (reference location).

Figure 2A provides examples of neuron pairs. We considered pairs ranging from highly similar to orthogonal orientation preference, and from perfect spatial overlap to a center-to-center separation of three times the receptive field (RF) size (Fig. 2B, left). The likelihood ratios for three example orientation differences (10°, 50°, and 90°) across 81 locations are shown in Fig. 2B. The complete set of 9 orientation differences is depicted in Supplementary Fig. 1. The 2D likelihood maps reveal that the shared modulator model largely outperforms the independent modulator model when two neurons have overlapping receptive fields and similar orientations (as shown in Fig. 2B, where green squares occupy a much larger portion of the grid on the left compared to the right). Conversely, the independent GSM has a higher likelihood with non-overlapping RFs with different orientations, although the numerical difference appears less prominent (Fig. 2B, right).

### Pairwise models and image statistics predict when surround stimulation suppresses or facilitates correlations

We next examined the predictions of the GSM models for pairwise neural activity. First, we analyzed the shared and independent GSM models applied to an example natural image windowed either by a small or large aperture (i.e., at two sizes). To illustrate the shared GSM, we considered a pair of model neurons with overlapping RFs ($\Delta x = 0$ and $\Delta y = 0$; Fig. 3A inset) with different orientation preferences ($\Delta\theta = 40°$). For the independent GSM, we considered a non-overlapping pair ($\Delta x = 2.25 \times RF$ and $\Delta y = 0$; Fig. 3C inset) with identical orientation preferences to those of the shared GSM.

We next asked if the covariability between neuron pairs differs for small and large images, focusing on correlations (often referred to as spike count correlations, noise correlations, or $r_{sc}$, which measure the Pearson correlation of spike count responses across repeated identical stimuli) as is commonly done to measure changes in covariability beyond those due to changes in single-neuron variance. The shared and independent models exhibited opposite effects of surround modulation on correlations. Specifically, increasing stimulus size decreased correlations from 0.96 to 0.78 in the shared model (Fig. 3B), but increased it from 0.08 to 0.28 in the independent model (Fig. 3D; see Discussion and Supplementary Fig. 2 for considerations about the magnitude of correlations in the simulations versus typical V1 data).

The opposite contextual modulation effects in the models stem from the different sources of uncertainty about the latent features $g_{c1}$ and $g_{c2}$. This uncertainty is determined by both the global modulators and additive noise, as depicted in Fig. 3E, F. In the shared model, uncertainty about the global modulator is the main source of shared variability among neurons. In contrast, in the independent model, distinct global modulators induce private variability for each neuron. In both models, the input noise (depicted in the brown sections of Fig. 3E, F) contributes to shared variability simply reflecting overlap between filters (see Methods). Importantly, the contribution of the additive noise to variability is not affected by stimulus size (Supplementary Fig. 3). As stimulus size increases, the shared model exhibits reduced uncertainty linked to the shared modulator, thereby decreasing shared variability and, consequently, correlations. Conversely, the independent model typically shows a decrease in independent variability, thereby allowing the other source of correlations (i.e., the additive noise) to become more evident. The schematics in Fig. 3E, F illustrate how changing the image size affects uncertainty regarding the latent features. In Supplementary Fig. 4, we demonstrate this intuition more formally, to show

that these contextual modulations of uncertainty result from marginalization of the global modulators (which is required for correct probabilistic inference of the $g$ latent features) and are absent when marginalization is neglected.

Having illustrated the effects for one example image, we then simulated responses of neuron pairs with varying degrees of tuning similarity, to a diverse set of 500 natural images in the BSD500 image set[43] (a subset of these images were used in the experimental recordings), distinct from those used to train the GSMs. We found that correlations depended on tuning similarity in both models, as has often been reported in V1[7,44–46]. However, the modulation by large stimuli was opposite for the two models, showing primarily suppression in the shared modulator, and primarily facilitation in the independent modulator. Additionally, the modulation was stronger for pairs with more similar tuning. Figure 3G demonstrates these effects for three example images (see Supplementary Fig. 5 for more example images), and in aggregate across 500 images.

### Surround stimulation modulates correlations in macaque V1 consistent with GSM predictions

We tested the predictions derived above, in V1 neuronal population responses recorded in anesthetized macaque monkeys. According to our theory, the responses of a neuron pair should sample the posterior of the better model of image statistics for that pair. If the visual inputs are best captured by the shared modulator model, increasing image size should reduce correlations. If inputs are best captured by the independent modulator, increasing image size should result in stronger (more positive) correlations (Fig. 3).

Because our analysis of image statistics indicated that the distance between RFs of the model neurons is the primary factor determining which model better captures input statistics (Fig. 2), we assigned the recorded neurons into two groups, depending on their RFs distance from the center of the stimulus (details in Methods). The first group, termed *centered* neurons, encompassed neurons whose RFs overlap the stimulus (distance between RF center and stimulus center < 1°). The second group, termed *off-centered*, comprised neurons whose RF falls outside the small image but inside the large image area. According to our analysis of image statistics, pairs of two centered neurons (*centered* pairs, Fig. 4A, top-left; similar to the model neural pair exemplified in Fig. 3E) are expected to follow the shared modulator prediction, whereas pairs comprising one centered and one off-centered neuron (*mixed* pairs, Fig. 4A, bottom-left; similar to the model neural pair of Fig. 3F) should follow the independent modulator. Since we are studying the effects of surround modulation on correlations, we did not analyze the off-centered–off-centered pairs because they are not driven by small stimuli.

Figure 4A displays findings from one example session. In this experiment, 270 natural image patches in two sizes were used: one windowed to fit the average RF (1°) and the other extending to the RF surround (6.7°). In centered pairs, we observed significant suppression of correlations by the larger image ($n_{cent} = 50$ neurons, ncase = 41,043 pairs and images; mean correlations: small images, 0.15; large images, 0.10; $p < 0.001$), whereas in mixed pairs there was significant facilitation of correlations ($n_{cent} = 50$ neurons, $n_{offcent} = 49$ neurons, ncase = 5654 pairs and images; mean correlations: small images, 0.04; large images, 0.08; $p < 0.001$).

This verified that the results for example session were a robust feature of V1 correlations (Fig. 4B). We analyzed data recorded with planar arrays (Utah) across 8 sessions in three animals, and one additional session using Neuropixels. Planar arrays allowed us to study multiple diverse combinations of spatial RFs, whereas Neuropixels afforded greater control in defining centered and off-centered neurons. In the aggregate data set we observed suppression of correlations by larger images for centered pairs (mean correlations and standard errors: small images, 0.1088 and $1.6 \times 10^{-4}$; large images,

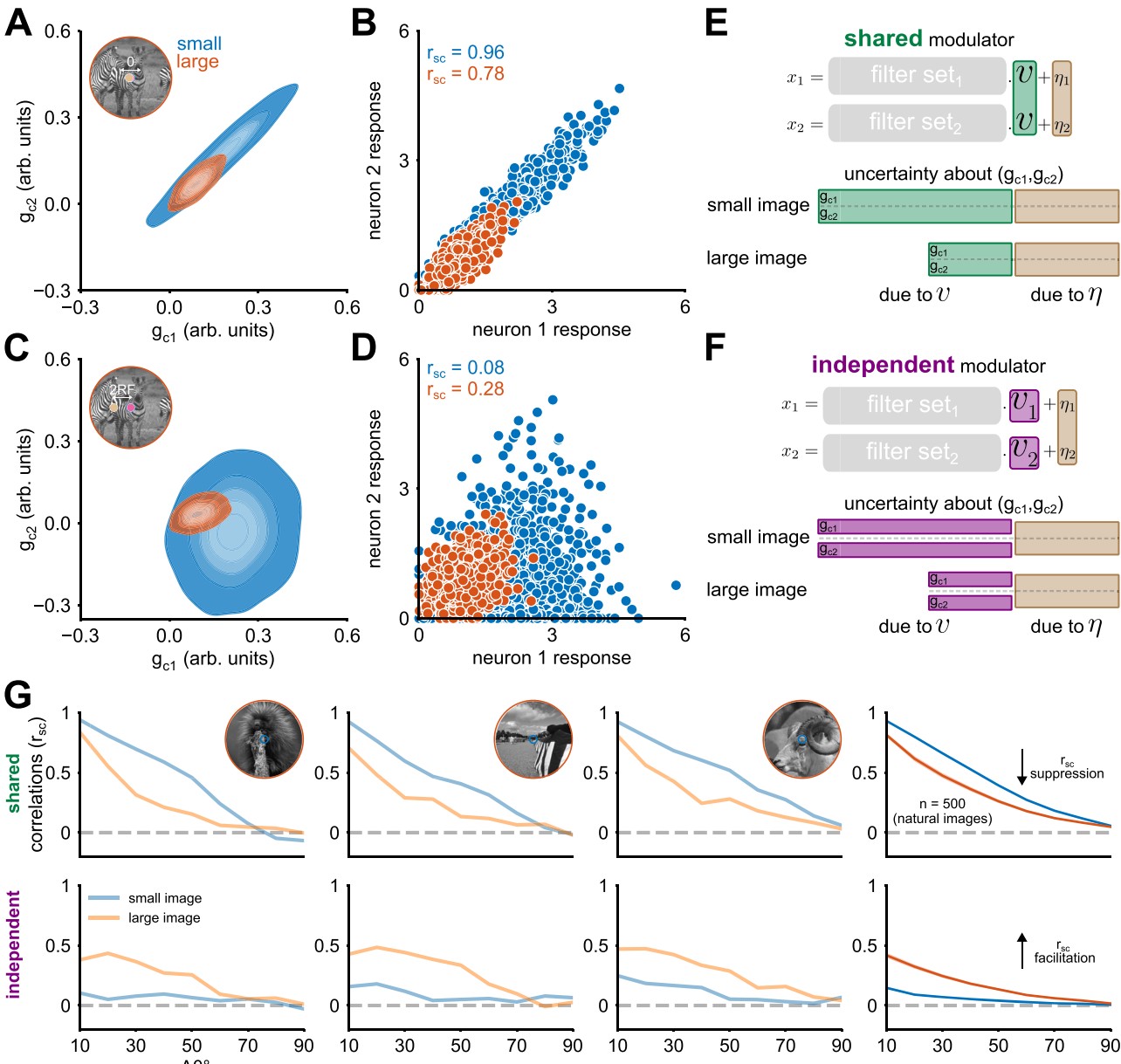

**Fig. 3 | Different GSM structures predict surround suppression and facilitation of correlations. A**, **C** Joint posterior distributions of the features ($g_{c1}$ and $g_{c2}$) encoded by a pair of neurons for one example natural image. The contours represent isoprobability regions at levels of 0.05, 0.25, 0.45, 0.65, and 0.85. Surround modulation decreases the posterior covariance for large images (orange) in the shared modulator model (**A**) and increases it in the independent modulator (**C**). **B**, **D** Neuronal responses are a transformed version of samples from the posterior distributions (details in Methods). Responses were simulated across 1000 trials. Trial-by-trial covariance reflect the posterior covariance of $g_{c1}$ and $g_{c2}$. **E**, **F** The schematic highlights two primary sources of covariability: the global modulator and the input noise. In both models, the contribution of input noise does not change between small and large images (brown). However, the uncertainty associated with the global modulator decreases with the addition of image context in larger images. Consequently, this reduction decreases the shared variability ($v$) in the shared modulator model (**E**), but it decreases the independent variability ($v_1$ and $v_2$) in the independent modulator model (**F**). **G** Correlation coefficient ($r_{sc}$) between two model neurons, calculated from 2000 trials (i.e., samples from the posterior). Top row: shared modulator for neural pairs with perfectly overlapping RFs; bottom row: independent modulator for neural pairs with non-overlapping center pairs (consistent with the statistics of natural images, see Fig. 2B). The first three columns from the left show $r_{sc}$ for three example images, and the rightmost column averages across 500 images (shaded areas correspond to the 99% confidence intervals). Blue and orange lines correspond to small and large images, respectively. See also Supplementary Fig. 2 for how the effects depend on the parameters of the GSM models.

0.0885 and $1.44 \times 10^{-4}$; $p < 0.001$) and facilitation for mixed pairs (mean correlations and standard errors: small images, 0.0472 and $4.5 \times 10^{-4}$; large images, 0.0515 and $3.5 \times 10^{-4}$; $p < 0.001$). The same result held in each one of the individual sessions (Supplementary Figs. 6 and 7). We verified the robustness of our results to changes in the thresholds defining centered and mixed pairs (Supplementary Figs. 6, 7, 8, 9). Lastly, we confirmed that the suppression and

facilitation effects were statistically significant on an image-by-image basis (Supplementary Fig. 10).

We were concerned that the modulation of correlations might simply follow firing rate effects, given the well-known downward estimation bias of correlations at low spike counts (see[7] for review). Specifically, for centered pairs, the suppression of correlations might reflect that firing rates decrease with larger images (Supplementary

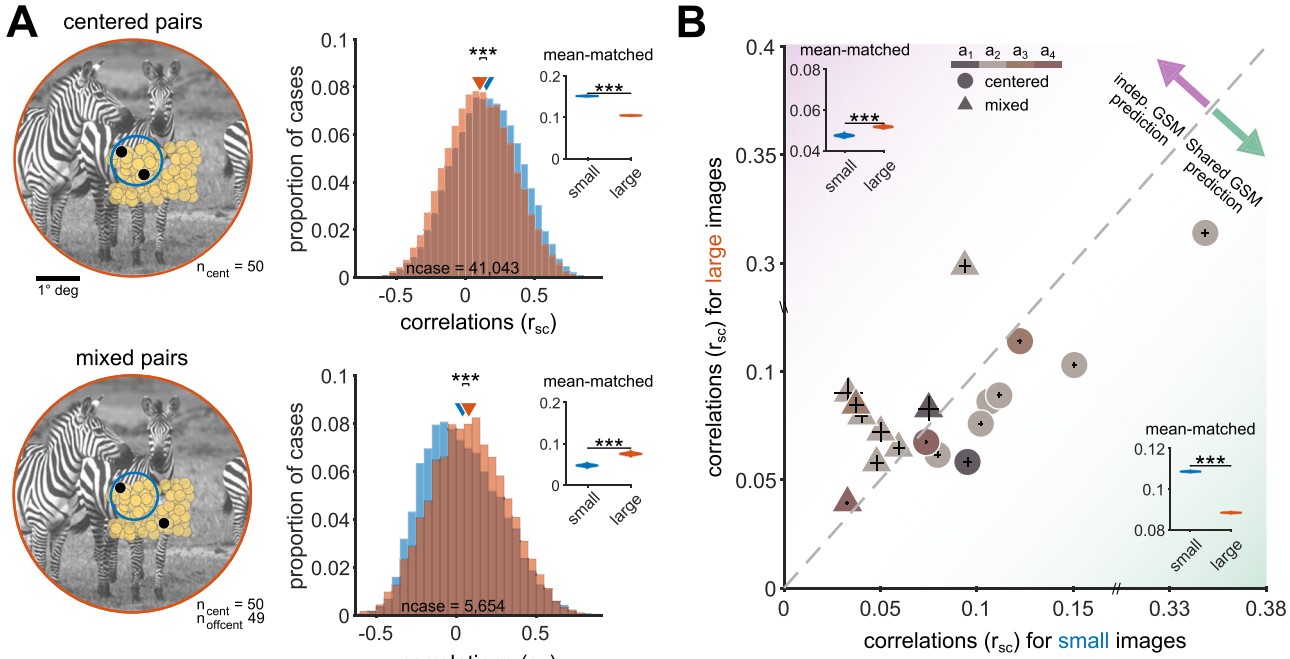

**Fig. 4 | Surround modulation of correlations in V1 aligns with model predictions. A** Left column: Light yellow circles depict the centers of the neurons' receptive fields in one recording session with a Utah array; black circles highlight one example `centered' pair. Small images (blue circle) were presented at 1° and large images (orange circle) extended across 6.7°. $n_{cent}$ and $n_{offcent}$ denote the total number of neurons centered on the stimulus or offset by more than 1.2°. Top row: black circles indicate a `centered' pair (i.e., both neurons are centered on the image); bottom row: black circles indicate a `mixed' pair (i.e., one neuron is centered on the image, the other is offset). Right column: correlations distributions for small (blue) and large (orange) images of centered pairs (top) and mixed pairs (bottom). The triangles represent the means of the distributions. For centered pairs (top), the mean correlation for small images significantly exceeds that for large images ($p < 0.001$; two-sided $t$-test against the null hypothesis of no difference). For mixed pairs (bottom) the mean for large images is significantly higher than for small

images ($p < 0.001$; two-sided $t$-test against the null hypothesis of no difference). To ensure that observed differences in correlations were not influenced by varying spike counts between small and large images, we specifically analyzed cases with mean-matched spike count distributions. In both rows, the inset illustrates the distribution of bootstrapped mean $r_{sc}$ following mean-matching analysis. **B** Each symbol represents the average $r_{sc}$ across cases (pairs and images) in a recording session. Circles represent centered pairs, while triangles indicate mixed pairs. Colors correspond to the results from four animals ($a_1$ to $a_4$) across nine sessions. See Supplementary Table 1 for detailed session information. The error bars represent the standard error of the mean. The insets show the distribution of bootstrapped mean $r_{sc}$ from a mean-matching analysis conducted across all sessions. The background shading indicates the predictions of the shared (green) or independent (purple) GSM models.

Fig. 11). For mixed pairs, enhanced correlations might reflect that the off-center neurons are not driven by small images, and so their firing rate increases substantially with large images. However, for these mixed pairs, large images also generally suppress the firing rate of the centered neurons (Supplementary Fig. 11, and ref. 2), resulting in a weaker net effect on the firing rate of the pair.

Nevertheless, to test whether the modulation of correlations was a trivial consequence of altered responsivity[47], we conducted a mean-matching analysis. We computed mean spike counts averaged across trials and across each neural pair, for each image and size. We then constructed two histograms across images, separately for small and large sizes. Finally, we resampled those histograms so that the mean for the small image matched the mean for the large image; see Methods. This analysis indicated that the observed correlations change were not due to differences in spike count means (Fig. 4, insets; and Supplementary Fig. 12).

### Surround modulation of correlations depends on tuning similarity and pairwise distance

Model simulations revealed a direct relationship between correlations and the similarity of the orientation preference of the two neurons, for both shared and independent GSM (Fig. 3). This is consistent with the well-known empirical relation between tuning similarity (sometimes termed signal correlation or $r_{signal}$) and correlations, typically measured with simple stimuli[45]. We confirmed that a similar relationship exists in our data, even when we measured tuning dissimilarity (i.e., 1 -

tuning similarity) based on responses to natural images (details in Methods; Fig. 5A, left for one example session and Supplementary Fig. 13 for all sessions). Consistent with previous observations[45], we found an inverse relation between correlations and RF distance (Fig. 5A, right). These relationships held separately for large and small images and for centered and mixed pairs. Additionally, similar to our simulations, the mixed pairs had lower correlations than the centered pairs across both small and large images, a difference likely due to the different inter-neuron distances among the centered and mixed groups.

With the broad range of RF distances and tuning similarities measured, we can exhaustively test how surround modulation of correlations depends on these parameters. Specifically, we studied the relationship between surround modulation of correlations, $r_{signal}$, and distance in the pairwise models and V1 data (Fig. 6). We binned the pairs by $r_{signal}$ computed from natural images and the proximity of their receptive fields. In each bin, we determined if the shared or independent GSM captured natural image statistics better (as in Fig. 2), and computed the modulation of correlations using the best model per bin (Fig. 6A, B). Figure 6B illustrates the relationship between correlations, $r_{signal}$, and distance, spanning 127,500 instances (17 distances, 15 differences in orientation preference, and 500 natural images). Neuron pairs with overlapping filters and high response similarity showed suppressed correlations, while those with non-overlapping filters and lower response similarity exhibited enhanced correlations. The shared and independent models alone do

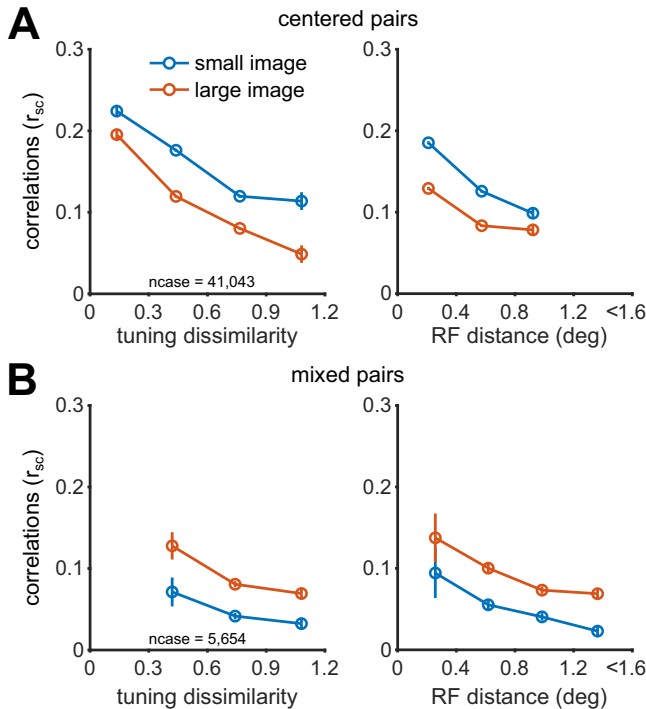

**Fig. 5 | Correlations are modulated by both tuning similarity and the RF distance between pairs of neurons. A** Centered pairs. Left: pairs were binned by their tuning dissimilarity (equal-sized bins). Circles denote average correlations (ordinate) and average tuning dissimilarity (abscissa) per bin, across small images (blue) or large images (orange). Error bars denote s.e.m. Right: pairs were binned by the distance between the RF center of the two neurons. Same plotting convention as in the left panel. **B** Mixed pairs. Same plotting conventions as in (**A**).

not account for the observed shifts between suppression and facilitation of correlations in the data (Supplementary Figs. 14 and 15).

Figure 6C outlines the relationship between correlations, $r_{signal}$, and distance based on V1 data from all recording sessions (354,498 pairs and images). A linear regression model with the z-scores of $r_{signal}$ and distance as predictors, explained approximately 38.3% of the variance in the modulation of correlations by image size (R-squared = 0.383). The linear regression coefficients were 0.15 ($p = 0.027$) for $r_{signal}$ and −0.57 ($p < 0.001$) for distance (see Methods for details). This analysis indicates that distance influences the modulation of correlations more strongly than tuning dissimilarity: correlations were on average suppressed by larger images for most neuron pairs with receptive field distance within 1° (blue pairs), and enhanced for more distant pairs (red regions). The effect of tuning dissimilarity was weaker but also notable, with pairs exhibiting high tuning similarity experiencing greater modulation of correlations by image size. Subsequent analyses with reduced models, excluding either $r_{signal}$ or distance, underscored their respective contributions. Omitting distance decreased R-squared by 0.3, emphasizing its substantial influence on the modulation of correlations. Conversely, excluding $r_{signal}$ resulted in a smaller reduction in R-squared by 0.02, indicating a lower, albeit significant, impact.

In summary, in both the model and V1 data, RF distance influenced surround modulation of correlations more than tuning similarity. In particular, the pairwise GSM with a global modulator variable shared between features at short distances−centered pairs in V1−suppresses correlations. Whereas independent modulator variables are a better model of V1 correlations for mixed pairs.

## Discussion
We have proposed a normative theory of V1 encoding of natural visual inputs, and empirically tested predictions for modulation of V1

covariability by spatial context (as manipulated by image size). We provide two main contributions. First, our work substantially extends the theory of neural sampling in the GSM model, leading to a new prediction for surround modulation of covariability. Specifically, our generative models of image statistics predict that surround stimuli reduce shared uncertainty, and thus suppress covariablity, for pairs of neurons with spatially overlapping RFs and similar tuning. Conversely, surround modulation strengthens covariability for neurons with offset RFs (Figs. 2 and 3). Notably, these predictions are parameter-free, derived from a hypothesis regarding the computational goals of V1 populations and an analysis of image statistics. Our second contribution is an empirical test of these predictions in V1 responses to natural images. We find both surround suppression and facilitation of V1 correlations, depending on RF distance and tuning similarity as predicted by our theory (Figs. 4, 5, 6).

**Probabilistic inference calibrated to non-stationary image statistics requires diverse functional interactions in V1**
Prior work with GSMs showed that single-neuron V1 activity reflects probabilistic inferences about the image feature encoded by the neuron[24–28] and also captured some aspects of interactions between V1 neurons[24–26]. Our study goes substantially beyond that past work, through a detailed analysis of natural image statistics with surprising implications for V1 covariability.

By building explicit pairwise GSM models, we showed that simply extending the GSM to pairs of model neurons is not sufficient to capture the statistics of natural images. This is because the assumption made in those prior studies, that a shared global modulator variable scales the local features encoded by all neurons, breaks down when those features are sufficiently distant or different (Fig. 2). This implies that a GSM with independent modulators is a better generative model for neuron pairs encoding distant or different features. This conclusion is consistent with earlier work in computer vision[29,48] and computational neuroscience[22,31] capturing the non-stationary statistics of natural images: these comprise multiple homogeneous regions that are statistically different from each other (e.g., the textures corresponding to the fur of an animal and to the vegetation in the background).

This observation about image statistics led to the key new insight of this paper: because increasing image size reduces uncertainty due to the modulator, and thus reduces response variability in sampling-based representations[25,27,28], our pairwise models predicted opposite effects on covariability depending on whether the modulator is shared or not between neurons (Fig. 3). Previous V1 models most closely related to ours[24–26], may only capture the suppression of correlations by spatial context, not the facilitation, because those models assumed shared modulators only. We confirmed that these results require that the probabilistic inference about image features takes into account the modulators in the GSM (i.e., marginalization, see Methods; Supplementary Fig. 4).

Here we have assumed that a given pair of neurons will always follow the predictions of either the shared or independent GSM, based on the learned prior statistics of the inputs received by that pair. With this simplifying assumption, our model predictions hold on average across presentations of many natural images (Figs. 4 and 5). However, functional interactions in V1 could be more flexible, allowing for switching between shared and independent modulators on an image-by-image basis. This is supported by earlier work that showed how probabilistic mixtures of GSMs capture flexible surround modulation of single-neuron firing rate[2,23,31]. Extending our pairwise model to probabilistic mixture models could thus provide finer-grained predictions for V1 responses to individual images, and new insights into the features of visual inputs that control functional interactions in V1.

We note that there is a quantitative difference in magnitude of correlations and surround modulation, between the pairwise model and V1 data. These differences may stem from the diverse pool of

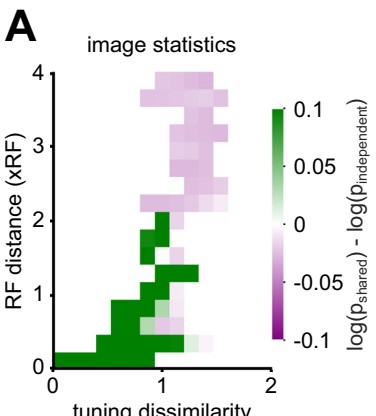

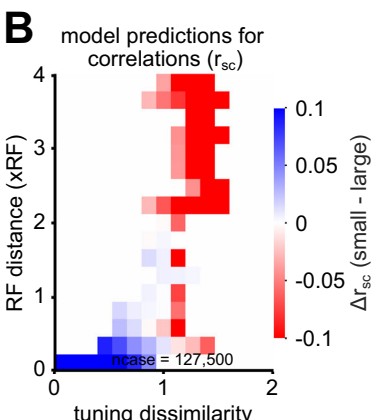

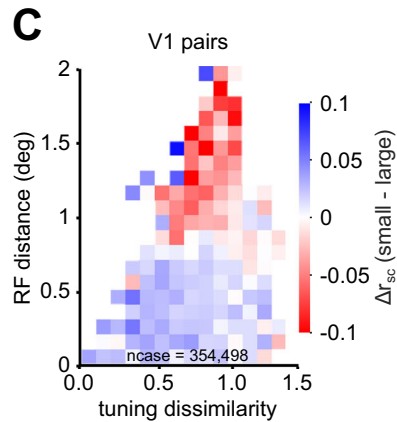

**Fig. 6 | Surround suppression or facilitation of V1 correlations is predicted by the optimal pairwise GSM structure for natural images. A** We binned model neuron pairs by their signal correlations and RF distances (center-to-center distance), and computed the average log-likelihood ratio per bin for shared versus independent pairwise GSM across 10,000 natural images from the ImageNet test set (see Methods for details). The green (purple) entries indicate conditions where the shared (independent) GSM model is the better model of image statistics on average. We observe a sharp transition at specific RF distances, where the overlap between the centers and surrounds of two model neuron filters is reduced. **B** We binned model neuron pairs as in (**A**) and computed the modulation of correlations per bin for 500 natural images, i.e., the difference in $r_{sc}$ for small images minus large images. For pairs with overlapping receptive fields and high tuning similarity, correlations are often suppressed (blue). In contrast, for pairs with non-overlapping receptive fields and low tuning similarity, correlations are often facilitated (red). **C** We binned V1 neural pairs and computed the modulation of correlations per bin as in panel (**B**). Only pairs with at least one neuron RF centered on the stimulus were included (see Methods). Correlations were on average suppressed when neurons RFs are separated by less than 1° (blue) and facilitated otherwise (red), consistent with the model prediction.

tuning preferences in V1 compared to the more limited range in the pairwise model. Additionally, the scaling of shared additive noise in the model, $\eta$, is arbitrary and could be adjusted to match the V1 data better. Here our primary goal was not quantitative model fitting, but rather to generate and test a qualitative normative prediction. We verified that altering the scale of shared additive noise separately for overlapping and non-overlapping pairs did not affect the qualitative prediction, though it could influence the magnitude of correlations (Supplementary Fig. 2).

### Relation to stochastic divisive normalization and implications for circuit mechanisms of V1 covariability

Our GSM model, and those of others, is closely related to divisive normalization[49]. Due to the multiplicative structure of the GSM, inference involves division of neural responses by an estimate of the modulator variable[2]. Importantly, when the modulator is shared between two neurons, their denominators are correlated, but when the modulators are independent the denominators are uncorrelated. Descriptive models of stochastic divisive normalization[28,50,51], when extended to pairwise data[52], indicate that normalization generally suppresses correlations when the normalization signals are correlated, whereas it enhances correlations otherwise. Therefore, our observations could be described by how surround stimuli recruit normalization signals with different properties depending on the relationship between the neurons' RFs.

The relation to normalization also points to an avenue to study the mechanisms implementing the probabilistic inference we have proposed. In particular, two related but distinct recurrent circuit models of V1 dynamics capture normalization. The supralinear stabilized network (SSN) captures key phenomena attributed to normalization[28] including surround modulation[53]. In stochastic versions of the SSN, recurrent dynamics shapes the noise[54], and the recurrence can be tuned to modulate variability as required by probabilistic inference in the GSM[26]. The ORGaNICs architecture[55] is designed to implement divisive normalization exactly at the steady state, and its stochastic variants also capture modulations of variability[56]. It is plausible that image-computable versions of both frameworks will capture the surround suppression of covariability that we observe here for overlapping RFs. The facilitation we observe for non-overlapping pairs may require additional tuning of the recurrent connectivity, to generate ensembles of neurons that effectively share normalization signals within each ensemble but not across ensembles. Lastly, as noted above, it is possible that interactions between a given pair of neurons could flexibly switch from shared to independent modulators depending on the visual input. We speculate that such flexibility might be achieved by feedback processes that dynamically refine the tuning of recurrent interactions based on the image context.

### Modulation of correlations by other stimulus features and attention

We have focused on spatial context in natural images because it has a prominent role in understanding the relation between image statistics and V1 encoding, and so it offers a strong test of our theory. Other stimulus factors also modulate correlations including, notably, stimulus contrast[44]. Past work has shown that inference in the GSM predicts contrast tuning of firing rate and quenching of variability[28], and lower correlations at higher contrast for overlapping RFs[26]. Experimentally, there is also a well-known interaction between stimulus contrast and surround suppression, namely reduced surround suppression at lower contrast[36,37]. This has been explained by the flexible engagement of the surround in a probabilistic mixture of GSM models[23]. If a similar flexibility affects also pairwise interactions, the modulations of correlations we have reported here may be reduced in magnitude at low contrast. This prediction remains to be tested.

Another study conceptually related to ours invoked probabilistic inference to understand pairwise V1 response statistics to synthetic textures and natural images[34]. They showed that patterns of correlations depend on high-level statistics, more than on low-level statistics. They explained the finding as reflecting that probabilistic inferences about high-level features (computed by higher visual cortex and fed back to V1) set the context for the inferences in V1. Different from our work, ref. 34 did not model natural image statistics or pairwise neural responses explicitly, and therefore did not make detailed predictions about what aspects of images enhance or suppress correlations. Trainable models for hierarchical inference[35] that reproduce the data of ref. 34 could be applied to the stimulus manipulations we have

considered. More broadly, other studies also support the view that perceptual inferences about task-relevant latent variables jointly modulate V1 responses, including correlations, via feedback[32,33,57–59]. Therefore, in addition to the recurrent mechanisms we discussed above, top-down feedback could contribute to the diverse modulation of V1 correlations by image size.

Endogenous attention is also known to modulate correlations in the visual cortex. Seminal studies reported primarily suppression of correlations by attention[12,60]. Subsequent work observed also facilitation depending on where attention is directed and whether the neurons provide evidence for the same or different perceptual choices[61]. Interestingly, stochastic divisive normalization has also been used to describe the diversity of attentional effects[62], similar to our proposal, although they did not offer a normative theory as we have done.

Despite these similarities, the theory we have developed is not directly applicable to attentional modulation, because the GSM models we considered includes only factors related to the visual inputs. In other words, in our models, neural variability encodes uncertainty and uncertainty reflects exclusively the local and global latent variables of the generative model of images.

Our experimental data were collected from anesthetized monkeys, to minimize eye movements and ensure more stable retinal input across trials, reducing stimulus-induced variability. It is possible that the effects reported here might differ in awake animals, where attentional fluctuations and feedback may influence correlations and center-surround modulation. We note however, that surround modulation of single-neuron response mean and variability with natural images is qualitatively similar and consistent with GSM predictions in both awake fixating and anesthetized animals[27].

### Implications for population-level functional interactions

We have focused on pairwise interactions, and our analysis of image statistics and correlations versus distance is relative to a reference RF location. An interesting direction for future work is to extend our modeling to populations of RFs that uniformly cover a larger area of the visual field. Our finding that neighboring RFs share the same modulator, and distant RFs use independent modulators, has several implications for larger, spatially distributed populations. First, to satisfy the constraint posed by pairwise image statistics, RFs should be organized into spatially compact ensembles, sharing the modulator within but not across ensembles. Second, more than two modulators would be necessary if spatially disconnected ensembles use independent modulators. Third, the coordination of ensembles by sharing latent modulators could offer a functional explanation for the widely observed low-dimensional population activity (assuming a smaller number of modulators than neurons). Fourth, this clustered organization would act as a strong spatial prior for image segmentation.

## Methods

### The pairwise Gaussian scale mixture (GSM) generative model

Our pairwise model adopts the neural sampling hypothesis as outlined by ref. 25: the instantaneous activity of two neurons represents samples from the joint posterior distribution of the features they encode. Thus, the covariability between neurons reflects the statistical dependence between those latent features.

To calculate the joint posterior distribution for a visual stimulus, we extended the Gaussian Scale Mixture (GSM) model—originally used to explain the statistics of single neuron responses[23,25,27]—to pairs of neurons. Specifically, our starting point was the model previously developed for surround modulation. We defined the observable variables by applying a set of oriented filters (similar to V1 receptive fields, RF) to grayscale circular image patches. The input vector, $x$, has 36 dimensions, corresponding to two groups of 18 filters, each group representing one model neuron. These filters, designed based on the steerable pyramid decomposition of the image[63], cover both the center and the surround of each neuron's RF. The 18 filters in each set share a common orientation, and are further divided into two subsets of 9 filters with even and odd phases, respectively, forming a quadrature pair. The nine filters in each subset include one representing the center of the neuron's RF and eight uniformly distributed around the center, representing the RF's surround (see Fig. 1A). Our results are robust across different filter scales by using multiple levels of the steerable pyramid (Supplementary Fig. 16). The motivation behind choosing common orientations for center and surround filters is twofold. First, several studies[64–66] demonstrated that neurons in V1 exhibit enhanced modulation when stimuli inside the RF and in the surround are similarly oriented, indicating orientation-tuned surround modulation. Second, we adopted this arrangement in our previous single-neuron model to capture the tuning of response variability[27]. Thus, our choice ensures that the model we use here captures known single-neuron surround modulation of both mean spike count and response variability (Supplementary Fig. 17). Nonetheless, because altering the orientation preferences of the surround filters can lead to different degrees of modulation depending on the structure of the visual input, we conducted additional simulations to verify the generality of our results: when averaging across a variety of natural images, the qualitative predictions for how correlations are modulated by stimulus size remain consistent regardless of the tuning of surround filters (Supplementary Fig. 18). We assumed that all surround neurons have identical, translated receptive fields. While fitting the receptive fields to images would not substantially alter our predictions—since the GSM framework captures the key statistical dependencies in natural images—we acknowledge that this constraint may limit the diversity of receptive field properties represented and thus the quantitative match to the data.

To infer local latent features, $g$, from an observation $x$, our model neurons invert the generative process of the GSM. In the shared GSM, the generative process involves the product of a global modulator $v$ that influences all local features $g$ and the addition of Gaussian noise $\eta$:

$$
\begin{aligned}
x_{ij} &= v g_{ij} + \eta_{ij}, & g &\sim \mathcal{N}\left(0, \Sigma_g^{\text{shared}}\right), \\
i &\in \{1, 2\}, & v &\sim \text{Weibull}(2, \sqrt{2}), \\
j &\in \{1, \ldots, 18\} & \eta &\sim \mathcal{N}(0, \Sigma_{\text{noise}})
\end{aligned}
\tag{1}
$$

where $i$ and $j$ index the neuron and the filter, respectively.

In the independent GSM, there are two independent modulators, $v_1$ and $v_2$, each scaling the 18 local features of the corresponding neuron:

$$
\begin{aligned}
x_{ij} &= v_i g_{ij} + \eta_{ij}, & g &\sim \mathcal{N}\left(0, \Sigma_g^{\text{independent}}\right), \\
i &\in \{1, 2\}, & v_i &\sim \text{Weibull}(2, \sqrt{2}), \\
j &\in \{1, \ldots, 18\} & \eta &\sim \mathcal{N}(0, \Sigma_{\text{noise}})
\end{aligned}
\tag{2}
$$

We assumed as usual in the GSM, that both $g$ and $\eta$ are multivariate normal variables. Their distributions have a mean of 0 and are characterized by covariance matrices $\Sigma_g$ for $g$ and $\Sigma_{\text{noise}}$ for $\eta$. In the independent GSM, the covariance $\Sigma_g^{\text{independent}}$ is assumed block-diagonal, i.e., no prior correlations between the filters of the two neurons (although there can be correlations between the filters representing each individual neuron). Conversely, $\Sigma_g^{\text{shared}}$ is not assumed to have block-diagonal structure. The variable $v$ is assumed to follow a Weibull distribution, with its scale and shape parameters set to 2 and $\sqrt{2}$, respectively. This choice is equivalent to the Rayleigh prior used in ref. 31, but is more readily implemented in the sampler (detailed in the next subsection). The multiplicative interaction between $g$ and $v$ implies that changes to the Weibull parameters would be equivalent to modifying the scale of $\Sigma_g$.

## Model training details

To find the noise covariance matrix ($\Sigma_{noise}$), we generated 10,000 synthetic white noise images. Starting with a normal Gaussian distribution, we shifted and scaled the values to range from zero to one, targeting a mean of approximately 0.5 and a standard deviation of 0.1.

Next, we applied two sets of filters, each corresponding to a model neuron, to the noise images and measured the empirical covariance among the filter outputs. In our model simulations, we introduced a free parameter to selectively adjust the covariance between these two filter sets. This adjustment preserved the variance and covariance within each filter set but scaled the covariance between the sets. By manipulating this parameter, we could change the levels of shared additive noise affecting the interaction between the two model neurons (Supplementary Fig. 2).

To estimate the covariance matrices $\Sigma_g^{shared}$ and $\Sigma_g^{independent}$, we considered 10,000 natural images: similarly to the noise images described above, we adjusted the pixel values of natural images to the range [0 1], and we then scaled them so that the signal-to-noise ratio between the natural images and noise images was 4.8. We then applied the filters to these natural images and we measured the covariance among the filter outputs to obtain $\Sigma_g^{shared}$. Next, $\Sigma_g^{independent}$ was derived by extracting covariances between each model neuron's filters from the full matrix $\Sigma_g^{shared}$ while setting the across-neuron blocks to zero (insets in Supplementary Fig. 19). The natural images were derived by manipulating 2500 natural images from the ILSVRC15 dataset[42]. As in ref. 2, each original image was rotated four times in 45-degree increments, to obtain similar empirical distributions of activations for filters of different orientations and improve numerical stability. While this reduces typical cardinal biases of natural images, it does not affect our qualitative results because here we addressed effects that depend only on the orientation difference between neurons, not on their absolute orientation preference.

## Relating probabilistic inference in the pairwise GSM to neural activity

We used Bayesian inference to estimate the posterior distribution of latent variables of the pairwise GSM. Specifically, because we assumed that neural activity represents samples from the posterior distribution over local features $\boldsymbol{g}$, our objective was to compute the posterior $p(\boldsymbol{g}|\text{stimulus})$, which involves marginalization over $v$ (note that we do not test, nor exclude, whether there is a circuit element, e.g., a neuron subtype or a specific circuit motif, tasked with explicitly representing inferences about $v$). There is no exact analytical solution for this distribution for the generative model with additive noise (Eq. (1), (2)). Therefore, we adopted the No-U-Turn Sampler[67], an extension of Hamiltonian Monte Carlo sampling, implemented via the PyMC3 probabilistic programming package[68].

We converted samples from the posterior into spike counts for model neurons following prior work on neural sampling[25,27]. We used a phase-invariant (i.e., complex cell) response model:

$$r = \alpha \left( \lfloor g_c^+ \rfloor_+ + \lfloor g_c^- \rfloor_+ \right) \qquad (3)$$

where $g_c^+$ and $g_c^-$ denote the latent features corresponding to the two filters with complementary phases in the center of the receptive field (RF). The notation $\lfloor . \rfloor_+$ represents the positive part of the response, rectifying the signal. In our analyses of model neuron responses we used $r$ as the instantaneous neural activity. As detailed in ref. 27 $r$ can be rounded directly to a spike count without additional spiking noise, as sampling-based models offer a normative explanation for variability. The coefficient $\alpha$ serves as a heuristic adjustment to ensure that the neuronal responses are kept within the range of experimentally measured spike counts. In Fig. 6, we introduce an offset to samples of $\boldsymbol{g}$ before converting them to spike counts. This adjustment helps mitigate the clipping effect caused by the rectifier,

particularly for smaller images, but does not change the qualitative effect (Supplementary Fig. 20).

To measure the covariability between two model neurons, we calculated the trial-by-trial correlation coefficient between the responses of two model neurons (denoted $r_{sc}$ in Figs. 3, 4). We used 2000 samples per input stimulus, drawn from the posterior across 4 independent chains (sequences in Markov Chain Monte Carlo simulations). Another 2000 samples are used for tuning before being discarded to adjust step sizes, scalings, etc. Employing multiple chains ensures a comprehensive examination of the parameter space and enhances the reliability of our Bayesian inference outcomes[68].

## Estimating likelihood of natural images in pairwise GSM models

To establish normative predictions about experimental covariability measurements, we assessed whether shared or independent GSM more accurately captures natural image statistics. This involved training the covariance matrices $\Sigma_g^{shared}$ and $\Sigma_g^{independent}$ with natural images, for neuron pairs with a large range of differences in spatial distance and orientation preference (Fig. 2). We then determined the likelihood of a test set of natural images (described above in Model training details) under each model configuration–shared or independent–for each filter set (using MLE covariance estimates yielding likelihoods consistent with moment-based results, Supplementary Fig. 21). The likelihood was computed as in[31].

By comparing the likelihood, we assessed which configuration of the GSM model, shared or independent, more accurately captured the underlying natural image statistics.

## Defining tuning similarity within the pairwise GSM model

To measure how correlations change with tuning similarity and distances in the pairwise GSM model for qualitative comparison with neural data (Fig. 6), we first assessed if the shared or independent GSM best capture image statistics for each pair of model neurons, and we measured the correlations modulation of the selected model. Next, we binned model neurons by the distance of their filters and by their tuning similarity (i.e., $r_{signal}$ measured across 100 natural images that stimulate the horizontal filter). Lastly, in each bin, we computed the average modulation of correlations. Using tuning similarity instead of orientation preference (as in Fig. 2) for direct qualitative comparison with V1 data accounted better for cases where two model neurons share an orientation preference but respond to different parts of the image.

## Animal preparation and data collection

We recorded from 4 anesthetized adult male monkeys (*Macaca fascicularis*), using established anesthesia and experimental protocols. In short, anesthesia was induced with ketamine (10 mg/kg), maintained with isoflurane (1.5–2.5% in 95% O2) during surgery, and switched to sufentanil (6–18 µg/kg per hour, adjusted as needed) for the recording session. We took several measures to ensure the stability and well-being of the subjects, including regulating the temperature and continuously monitoring vital signs such as EEG, ECG, blood pressure, end-tidal PCO2, and airway pressure. Vecuronium bromide (0.15 mg/kg per hour) was used to minimize eye movements. In the primary visual cortex (V1) of three animals, we implanted 10 × 10 multielectrode Utah arrays with 400 µm spacing and 1 mm length. A subset of the data we analyzed (7 sessions from two monkeys) have been reported in ref. 2.

In one animal, we used Neuropixel Phase 3B probes, secured on a custom 3D-printed holder. We used 4 sharpened probes, inserted without guide tubes. After-insertion, the craniotomy was sealed to preserve cortical integrity. The recordings, covering 384 channels across a 7.68 mm range, were made using SpikeGLX software. Spike sorting was performed with Kilosort 2.5, which clusters units based on waveform shape, and was manually refined with Phy software to ensure that the waveform belongs to single neurons.

The number of recorded units ranged from 22 to 53 for centered neurons and from 24 to 190 for off-centered neurons.

All procedures were approved by the Albert Einstein College of Medicine and followed the guidelines in the United States Public Health Service Guide for the Care and Use of Laboratory Animals.

## Visual stimuli and presentation

We used a calibrated CRT monitor and custom software for stimulus displays, featuring 1024 × 768 pixels resolution and a 100-Hz refresh rate, with a mean luminance of approximately 40 $cd/m^{-2}$. The monitor was positioned 110 cm away from the animal for Utah array recordings and 50 cm for Neuropixel recordings. To map the spatial receptive field (RF) of each neuron, we used small gratings (0.5° in diameter, in four orientations, presented for 250 ms) across various positions. The RF center for each neuron was identified as the peak location on a two-dimensional Gaussian fit to the spatial activity map[2,69].

Surround modulation was assessed using grayscale natural images, as outlined in prior research[2]. In summary, for 8 Utah array recording sessions, we displayed 270 distinct natural images (210 in one session) in two sizes (1° and 6.7°). Most images had a natural main orientation (defined by analyzing the histogram of orientation energy as in ref. [2]); those images were presented in four variants, each rotated in 45° increments, to increase the likelihood of activating the recorded neurons. Images were shown for 105 ms, followed by a 210 ms blank screen interval, in a pseudo-random sequence. Images were interleaved in pseudo-random order and each image was presented 20 times (seven sessions from ref. [2]) or 50 times (one session). Stimuli were displayed within a circular aperture against a gray backdrop that matched the average luminance.

For Neuropixel recordings, we chose a subset of the BSD500 images, which included 48 natural images presented at two sizes (1° and 6.4°). These images were presented in a pseudo-random order for 250 ms and each was repeated 150 times.

## Characterization of neuronal responses and inclusion criteria

We counted spikes in a time window as long as the image presentation, shifted by 50 ms (35 ms in one session) to account for typical response onset delays. Neurons were categorized into two groups: "centered" neurons whose receptive fields were within a 1° of the stimulus center, and "off-center" neurons whose receptive fields were more than 1.2° away from the stimulus center. For off-center neurons, we additionally ensured that the neurons' receptive fields were covered by the larger stimuli. Our results are robust to the specific definition of "centered" and "off-centered" (Supplementary Figs. 8 and 9).

We included in the analysis only the responsive neurons for each stimulus, as follows. We computed the baseline activity level as the spike count during spontaneous activity, averaged over all trials. We then identified neurons whose stimulus-evoked activity exceeded both one standard deviation above the baseline and a minimum value of 0.1 spikes/trial. Centered neurons were deemed responsive if they responded to small stimuli above the baseline threshold. Conversely, off-center neurons were deemed responsive if they responded to large stimuli above the threshold, and additionally did not respond to small stimuli above the baseline threshold to ensure the small stimulus did not encroach on their RF. In the Neuropixel data, we observed overall lower activity levels compared to the Utah array recordings. Therefore we lowered the responsivity threshold to 0.1 standard deviations above the spontaneous activity. Due to the absence of blank interval trials and the presence of stimulus-driven responses, we lowered the baseline threshold to a maximum of 0.1. Our findings did not change qualitatively when we changed the values of these thresholds, as shown in Supplementary Figs. 6, 7, 9.

To evaluate correlations (also known as noise correlations or spike count correlations, denoted by $r_{sc}$ in the figures), we computed Pearson correlation coefficients between pairs of neurons across trials (repeated presentations of the same image).

## Statistical analysis

In our analysis, we employed a two-sided t-test to determine if correlations significantly differed between small and large images. To avoid inflating significance due to the fact that the entries of an estimated noise covariance matrix are not independent, we proceeded as follows. In each session with $N$ recorded neurons, for each image we computed the correlation matrix for the $M_{image}$ neurons included for that particular image (see inclusion criteria above). We then randomly subsampled $M_{image} \times K$ pairs—where $K$ is the estimated dimensionality of the covariance matrix based on the low-rank approximation from factor analysis[70]—from the full set of $M_{image}(M_{image} - 1)/2$ possible neuron pairs. Lastly, we aggregated the sampled pairs across images and sessions to compute a $p$-value. This procedure was repeated 1000 times, and we report the average $p$-values across these repetitions. (See also Supplementary Table 1 for session-by-session significance).

To ensure that the observed differences in correlations were not simply due to differences in the average spike counts elicited by small versus large images, we conducted a mean-matching analysis, as follows. First, for each image and each size (small and large), we computed the across-trial mean response (spike count) for each neuron. Second, for each pair of neurons, we computed the average mean response of the pair, for each image and size. Third, we constructed the histograms of neural-pair-averaged mean responses, across all pairs and images, separately for small images and for large images. We then selected samples from these two histograms to ensure the means of the neural-pair-averaged responses were matching. Following this, we examined the correlations for these mean-matched cases. To evaluate the significance of any observed differences between the two groups, we applied a paired sample, two-sided t-test, based on the null hypothesis of no difference between them. We also confirmed that our qualitative results on surround modulation of correlations are unchanged when using a regularized covariance estimator[70] instead of mean-matching.

We employed linear regression to elucidate how surround modulation of correlations depends on tuning dissimilarity (1-$r_{signal}$) and proximity (distance) in Fig. 6. The analysis included 154 samples, selected from 300 bins each containing at least 15 data points, ensuring robust representation. All variables were z-scored to standardize measures, enabling comparability across different scales.

## Reporting summary

Further information on research design is available in the Nature Portfolio Reporting Summary linked to this article.

## Data availability

Data from seven sessions are publicly available through the CRCNS data sharing platform at https://crcns.org/data-sets/vc/pvc-8. Data from the remaining two sessions are available at https://doi.org/10.5281/zenodo.15596406. The natural images used to train and test the GSM models are publicly available in the ImageNet database (https://image-net.org/challenges/LSVRC/2015/) and the BSDS500 dataset (https://github.com/BIDS/BSDS500). Source data are provided with this paper.

## Code availability

Code used for model simulations and data analysis is available at https://github.com/CoenCagli-Lab/2025-NatureCommunications-Farzmahdi-et-al-code.

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

## Acknowledgements
We thank the R.C.C. and A.K. labs for insightful discussions, and the A.K. lab for help with experiments. We also thank Aravind Krishna for his assistance with data analysis and Dylan Festa for his help in modeling. This research was funded by the National Institutes of Health, with grants EY030578 and DA056400 awarded to R.C.C. The content of this publication is solely the responsibility of the authors and does not necessarily represent the official views of the National Institutes of Health.

## Author contributions
A.F., A.K. and R.C.C. designed the study; A.F. and R.C.C. developed the theory and models; A.K. and R.C.C. performed the experiments; A.F. analyzed the data; A.F., A.K. and R.C.C. wrote the paper.

## Competing interests
The authors declare no competing interests.
