## [Transparent Peer Review file · Nature Communications]

Relating natural image statistics to patterns of response covariability in macaque primary visual cortex

Corresponding Author: Dr Ruben Coen-Cagli

Version 0:

Reviewer comments:

Reviewer #1

(Remarks to the Author)

The paper presents a study on stimulus-dependence of noise correlations in V1. The paper introduces a model-based hypothesis on how noise correlations between pairs of neurons is modulated by changing the aperture in which natural stimuli are presented and this hypothesis is tested on multi-site electrophysiological measurements. The key insight of the paper is elegant and is based on a rather fundamental property of probabilistic modelling of visual processing: uncertainty in V1 is determined by multiple sources of variability some of these imply independent variability others carry shared variability; the authors point out that upon changes in stimulus presentation aperture predictable changes occur in the relative strengths of these sources of uncertainty, resulting in changes in measured noise correlations. This insight is both novel and important, and makes a significant contribution to population-level interpretation of the neural code, thus going beyond single-cell-level coding strategies. The experimental recordings are well aligned with the theory and provide a convincing case for the theoretical predictions.

I am raising two issues related to the theory/recordings and two additional issues related to the presentation.

First, the model consists of two 'focus' neurons that are flanked with a population of 'surround' neurons. These surround neurons feature similar sensitivities as the focus neuron, their receptive fields differ from the focus neuron only in the location of the RF center. This choice seems to be very specific but I could not find clear motivation for restricting analysis to such a model, and for retaining the generality of the model under the specific choice the authors made. Demonstrating the generality of the predictions of the theory would strengthen the paper. Also, it would be beneficial for the readers, what single-cell properties of the Gaussian Scale Mixtures model this simplified version retains.

Second, Figure 4 presents the experimental confirmation of the theoretical predictions. As far as I understand, the theoretical analysis assumes that the two focus neurons are both stimulated with different aperture images and it is the natural image statistics that predict disjunct modulators for the two focus neurons that results in the facilitation of noise correlations when focus neurons have distant RFs. In the case of the experiment, however, a slightly different scenario is presented: only one of the neurons is stimulated with the small aperture image, the other neuron has a grey image in its RF. This difference between theory and experiments is not evident in the text. Although the authors perform a large number of controls in which criteria for centered and mixed pairs are exhaustively explored, this specific distinction is not made explicit. I am wondering if a more direct test of the experimental setup can be performed with the model.

Comments on writing:

First, the paper introduces the Gaussian Scale Mixtures model as early as the introduction as a model of natural images. Then, the results introduces a very constrained version of the Gaussian Scale Mixtures model, that is not really a model of natural images but shares some characteristic features with the original model. I believe that a more careful transition to the model actually studied in the paper would be utterly beneficial. Without a proper motivation, the toy-looking model seems unconvincing as a model of natural images and the reader gets lost on the assumptions and approximations behind the model. This is especially important, as the discussion states that the theory is parameter-free as its parameters are fully determined by natural image statistics. While the statement is true, the model is also largely determined by the introduced simplifications, which could limit its generality and the reader needs guidance with respect to its generality.

Second, the paper makes important and exciting contributions to understanding pair-wise statistics of neural responses. To motivate the approach, the introduction explores earlier contributions. While the paper rightfully highlights that traditional approaches focus on single neuron responses, several works started addressed stimulus-dependence of noise correlations, which is central to interpret noise correlations as important components of coding. These studies include Lin (2015 Neuron), Franke (2016) Neuron, Banyai (2019) PNAS, Csikor (2024) biorxiv. In fact, latent variables that jointly modulate V1 responses and contribute to noise correlations has a wider literature, that spans papers such as Haefner (2016) Neuron, Lange & Haefner (2017) Curr Opin Neurobiol, Bondy (2018) Nat Neuro, Banyai (2019) Curr Opin Neurobiol.

Minor:

- * line 109: Bilal Haider's paper on the surround stimulation on reliability and sparseness seems especially relevant at this point
- * line 147: the notation is not introduced with sufficient care: the reader might easily get lost what it means
- * Fig 1: the step from posterior to sampling is indicated as inference, while the text refers to this step as representation of inference
- * lines 164—169: this statement seems very dense, I believe that some extra explanation would be welcome for the general reader
- * lines 181–183: training remains elusive here: the reader would benefit from some explanation what aspects of the model are trained
- * Fig 3a caption does not specify the top and bottom subpanels.
- * line 343: The mean matching procedure seems to be central to the findings. As far as the short description permits, the procedure is identical to the one described in Banyai (2019) PNAS. Based on Fig S12, the changes in noise correlations both for central and mixed pairs are similar to those that would be expected from de la Rocha (2007) therefore a direct comparison of the effect of aperture change *with* and *without* rate matching could be instructive. I applaud showing rate matched results but a convincing demonstration of the distinct effect of the theory and mere firing rate changes are important.
- * Refs Bawat et al, Schrimpf et al seem to be corrupted.
- * throughout the equations (eg lines 679 & 720): subscripts and superscripts ('shared') are not placed in $\mathrm{\{}}$.

(Remarks on code availability)

Reviewer #2

(Remarks to the Author)

Overview

At a high level, this paper (1) makes a novel observation about a qualitative phenomenon in an existing model family (the GSM), (2) formulates this as a (qualitative) prediction for pairwise correlations among neurons in V1 using an established framework (neural sampling), and (3) finds evidence for the predicted trends in new data collected from anesthetized macaque V1. Building bridges between theory and data in this way is challenging and important, and I think that the work here is high quality. However, I feel that the novelty of the work is overstated (some key references missing). It is also an unfortunate standard in this field that novel predictions are so often evaluated on novel data without (1) benchmarking the predictions on extant data, nor (2) benchmarking earlier models on the new data. Still, I think this paper contributes some genuinely novel and interesting insights on the relation between sampling (in a GSM) and noise correlations.

Major comments

1. The authors state in the introduction that predictions from the neural sampling hypothesis have been limited to marginal statistics (individual neurons' means and variances) without addressing pairwise statistics (covariances). This is not true; see, for instance, refs [1-4] below (none of which is currently cited).
2. Regarding model benchmarks, the core of this paper is a prediction about stimulus-dependent correlations in a variant of the well-studied GSM model. The authors note that other papers with other model variants in the GSM family have qualitatively reproduced a number of marginal statistics of V1 activity (e.g. contrast-dependence, tuning curves, variance and fano factor, etc.). Do all of those same marginal statistics still hold in the new model? I would argue that such sanity-checks are a necessary bar to clear before addressing novel predictions of the new model. (I don't expect this is true here, but one can in-principle imagine constructing a model that makes new and interesting predictions for pairwise neural statistics but fails to recapitulate basic facts about their marginal statistics).

Put more succinctly: for a new model M2 to supersede an earlier model M1, M2 should explain everything that M1 explains, and more. It is an unfortunate standard in our field that new models are so often tested on new data without replication or generalization tests across models/datasets.

Notably, the model by Orban et al (2016) was also a GSM but they used MAP inference over the modulator v . The present paper replaces this with joint inference over (v, g) . If my understanding is correct (see comment on Simpson's paradox in Minor point 1 below), then this simple modification of replacing MAP inference with full inference may be *the* key driver of the predictions about correlations. The authors may want to illustrate this explicitly, showing that MAP inference of v does *not* make the same predictions, which would be a major point in favor of the present work above and beyond the (otherwise very closely related) prior work by Orban et al.

3. Insufficient details in the experimental methods and reporting of differences between subjects.

It appears in Fig 4b that effects for the utah-array recordings and neuropixels recordings may be qualitatively different. It is unclear from Fig 4b which sessions correspond to which subject. It would be helpful to see more detail on individual subjects' recording sites, number of isolated units, putative differences in cell types or cortical layer, etc. This is especially crucial in any analysis of correlations, since correlation estimates can be contaminated by mislabeled spikes (in the neuropixels data) and by background fluctuations in the utah-array data [5].

4. Statistical concerns in data analysis.

Why was the mean-matching control necessary in the neural data (L938) if it was not necessary in the model (L780)? It's of course expected that mean firing rate will impact variance, but I do not follow the logic that it will impact the correlation estimates, unless the concern is about using a biased estimator of correlation, which could be bias-corrected a number of ways (and would also impact the model predictions)

The authors appear to use correlation values computed from the raw (sample) covariance matrices, which will in general give biased estimates of the true correlation [6]. The authors might be interested in using a regularized covariance estimator [7] without the mean-matching procedure. In particular, [7] includes a diagonal + rank-1 covariance estimator that seems appropriate in this setting where the hypothesis on "mean correlation" changes is essentially equivalent to hypothesizing a change in variance along the all-ones vector, i.e. a change in the variance of the population mean, without a commensurate change in other dimensions of the space.

I am wary of statistical tests on correlation that lump all $N(N-1)/2$ unique correlation values into a test as if they are independent samples from a population. If, say, the correlation effects are driven by a small set of latent modulators (as in the GSM model), then the number of degrees of freedom in the correlation matrices is on the order of N rather than N^2 . I suspect that this effect is inflating the reported significance of the results.

Given that the stimuli are presented in pairs, where the same image is shown small/large, this suggests to me that there is a stronger statistical test available: following the logic above, the "change in average r_{sc} " prediction is essentially a prediction for the change in variance of the population average activity, and no-change or less-increase in orthogonal directions. Could this be tested within-stimulus by comparing responses to small vs large versions of the same image?

Moderate/Minor comments

1. While the figures are generally very well done, I am concerned that the illustrations of correlated variability in Figs 1C and 3A are hiding a critical detail.

The short version: the figures, as drawn, make the posteriors appear Gaussian, but their non-Gaussianity (due to uncertainty about v) is a critical aspect of the theory. These sketches would be clearer if they showed contours of the true posterior.

The longer version: As I understand the predictions of the model, the predicted positive correlations among overlapping and similarly-tuned neurons are a case of Simpson's paradox. That is, for any given value of the modulator v , there is a negative correlation among the g 's (explaining-away); but, uncertainty over v drives coordinated changes in the posterior means of the g 's, resulting in a net positive correlation. (This could be done analytically by decomposing the posterior covariance into the average covariance conditioned on each v , and the covariance of the posterior means across different v 's). If this intuition is correct, it would be much clearer to illustrate the Simpson's paradox effect in the figures.

2. I am not sure how to think about the model constraint that all surround neurons have identical but translated projective fields. Would the same predictions not hold for a GSM in which the projective fields are also fit to images? Again in the spirit of model-benchmarking, I would have appreciated at least some discussion on what limitations (or not) there are due to this constraint.

3. I was surprised to see that the data were collected from anesthetized monkeys. I suggest the authors at least comment on whether/how they expect the results to change in awake subjects, especially given that anesthesia affects inter-area communication, and (as far as I'm aware) some center/surround effects are thought to be driven by feedback.

4. The model-fitting procedures seem non-standard. Why not use the EM algorithm for fitting the GSM (derived by the authors in 2009 work)? What is done here – using empirical covariances of filter responses to natural and noise stimuli – is perhaps an approximation to the MLE parameters (and therefore a good way to initialize EM), but not equivalent to the MLE parameters. It is also not obvious to me whether the extent of this approximation is worse in the shared vs independent models. Ultimately, this complicates the interpretation of the likelihood comparisons in Fig 2 (though it is unlikely to affect the qualitative trends).

References

1. Bányai, M. et al. Stimulus complexity shapes response correlations in primary visual cortex. Proceedings of the National Academy of Sciences 116, 2723–2732 (2019).

2. Bányai, M. & Orbán, G. Noise correlations and perceptual inference. *Current Opinion in Neurobiology* 58, 209–217 (2019).
3. Haefner, R. M., Berkes, P. & Fiser, J. Perceptual decision-making as probabilistic inference by neural sampling. arXiv (2014).
4. Lange, R. D. & Haefner, R. M. Task-induced neural covariability as a signature of approximate Bayesian learning and inference. *PLOS Computational Biology* 18, (2022).
5. Schulz, D. P. A., Sahani, M. & Carandini, M. Five key factors determining pairwise correlations in visual cortex. *Journal of Neurophysiology* 114, 1022–1033 (2015).
6. <https://stats.stackexchange.com/questions/148346/shrunken-r-vs-unbiased-r-estimators-of-rho>
7. Yatsenko, D. et al. Improved Estimation and Interpretation of Correlations in Neural Circuits. *PLoS Computational Biology* 11, 1–28 (2015).

(Remarks on code availability)

Reviewer #3

(Remarks to the Author)

The study by Farzmañhi et al contains important findings and insights regarding the nature of shared vs independent sources of variation and how such sources will affect the representation of natural images in V1. Overall, I like this study and I think that the paper has the potential to have a broad readership and impact. Toward that end, I have suggestions about how best to frame the work so that its importance becomes clear.

1. The first part of the paper describes in detail the GSM model, which the authors are known for developing. It is a useful model on its own terms, but I would respectfully suggest focusing specifically on the design feature involving the shared vs independent modulator parameter “v”. The details of the GSM except for this modulator parameter could be couched in very general terms and/or moved to the methods section, which would permit focusing on the central question of the global modulator vs separate modulator signals. Terminology in figures could be “shared modulator”, “independent modulator” etc, to focus the reader on that part of the simulations rather than on the GSM as a whole.
2. Doing so would allow the authors space to present the idea conceptually, which would (presumably) help alleviate one of my confusions – if the modulator is the only source of shared (or not) variability that is provided to the neurons in the modeling section, then how is it the case that there are any noise correlations at all in the independent modulator model (Figure 3A, B – bottom panels)? This led me to wonder if the analysis was mixing signal and noise correlations? Signal correlations get defined and used later in the paper so it seems like that’s not the case but then I’m left without an explanation for this finding.
3. Generally, both the figures and the text could use some attention at both the flow and detail levels. Specific examples provided below, but are not intended to be an exhaustive list. In particular, attention should be given to the abstract and introduction to ensure that they set up the study to help the reader understand what was done and why it matters.

Abstract:

“A distinctive prediction of our theory is that spatial context in images modulates noise correlations, and the sign of the modulation depends on the overlap between receptive fields.” Suggest changing “spatial context” to “image size” – this is a more specific description of what was tested as there isn’t any breakdown or discussion of the results in terms of specific types of images and how they might vary in terms of the scale over which one part of the image is correlated with another.

“This is because spatial context reduces shared uncertainty (a source of shared variability) about overlapping image features, but it reduces private uncertainty about non-overlapping features.” I don’t understand the phrasing of this sentence. Specifically, “context reduces shared uncertainty...but it reduces private uncertainty”. While part of the problem might be that one of the “reduces” should be “increases”, I think the whole sentence is a bit unclear, because it’s too distant from what was actually done in the paper. Suggest replacing with something that specifically describes the use of shared vs independent modulator signals and the impact on the results.

“Analysis of macaque V1 responses to natural images supports our theory, revealing that surround modulation switches from suppressing to enhancing noise correlations as the spatial offset between receptive fields grows.”. The terms “suppressing” and “enhancing” are not clear at this point of the paper because noise correlations can be either positive or negative. Would be better to stick close to the findings themselves – I assume this refers to figure 5, so something like “noise correlations are more positive for small images than large images in pairs of neurons with similar receptive field locations and orientation tuning preferences, but the noise correlations are more positive for larger vs smaller images when the two neurons have different receptive field locations and/or orientation tuning”. (That’s not a great suggestion; hopefully you can come up with a crisper way of putting it.)

Introduction:

I think the introduction would be more helpful to the reader if it arrived at the central idea of this study sooner. The shared vs. independent modulator signals don’t appear until we are about 3 paragraphs in to the Results section – I think this should come much earlier. To me, the key points the introduction should cover are: (1) we know that noise correlations matter to the important problem of inferring visual images from activity in V1; (2) with natural images, there are correlations in the images themselves across space that may be relevant to V1’s representation; (3) the brain might possibly be wired up to anticipate such correlations by having sources of modulation deployed across some spatial scale across the V1 representation; (4)

here we test a simple version of this idea using a previously established generative model configured in two particular ways, with shared vs. independent sources of modulation, and we test the impact of these two configurations on (a) how well image identity can be inferred from the V1 activity patterns (b) how noise correlations vary as a function of neuron pair and image size.

I'm sure that summary isn't exactly right! But I hope it's helpful way of seeing what this reader took away from your study.

Results:

"For the pairwise application considered here, we reasoned that joint inferences about pairs of features should take into account whether those features are a priori more likely to be part of the same or different visual objects," I like this insight a lot – can you say more? (perhaps not here, perhaps in the discussion?) How do you envision the brain knowing ahead of time whether two features are part of the same object and therefore deploying the same modulator?

Figure 1: the shared/independent modulator signal is a very tiny part of the figure – can this be enlarged to indicate that this is the key focus of the paper? Also, please stay away from yellow – the noise inputs were not visible in the left part of 1B. Also, my understanding is that the noise inputs are independent for each neuron, but the yellow boxes in 1B (right side) draw them as if they are shared.

Figure 2 and associated text: The purple/green panels in Fig 2B were difficult to follow because there appears to be some typo in the label for the color bar, with a "<" substituting for a "+"? Adding a common language description to the label would also help "Shared > Independent", "Independent > Shared" for example.

As a general rule, when describing figures, it can be very handy to say directly to the reader what exactly in the figure supports the point. There's an opportunity to do this in the text around line 200: "The 2D likelihood maps reveal that the shared GSM model largely outperforms the independent model when two neurons have overlapping RFs and similar orientations" could read: "The 2D likelihood maps reveal that the model with a shared modulator signal largely outperforms the model with an independent modulator signal when two neurons have overlapping RFs and similar orientations (green squares occupy a larger portion of the grid in Figure 2B left most panel vs a much smaller portion in the rightmost panel)".

Line 228: "rsc" as an abbreviation for noise correlations is first used here, I think. While this term is standard in the field, it would be very confusing here for anyone not steeped in this topic because "spike count" as the binding for the "sc" part is not used. I suggest switching to "rnoise" throughout (since later you have "rsignal").

Line 242: "In the shared model, uncertainty about the global modulator is the *main* source of shared variability among neurons". Isn't it the *only* source of shared variability among neurons, at least for repeated presentations of the same stimulus? If this is not the case, then it should be clarified earlier (I note that if I've missed this then that would account for my confusion about Figure 3 noted above.

"Centered" vs "off-centered" vs. "mixed" – are "off-centered" and "mixed" synonyms? Suggest picking one and sticking to it.

Page 5 – results in Figure 3 for simulations, results from V1 recordings – the magnitudes of the observed correlations are wildly different. This didn't particularly worry me, but might confuse other readers, so it might be worth a mention somewhere that the size of the correlations in the simulations is (presumably) a function of the experimenter-chosen values for the modulator signal.

Figure 4. "mixed pairs" used here, should this be "off-centered" pairs?

I hope these comments are helpful. I think the work is interesting and valuable.

(Remarks on code availability)

Version 1:

Reviewer comments:

Reviewer #1

(Remarks to the Author)

The authors have thoroughly and exhaustively addressed all my comments and I believe that with the proposed update of the manuscript is ripe for publication. I especially appreciate the novel insight that supplementary figure 18 provides (produced in response to Reviewer 2). I would encourage the authors to present this under the results as it shows a true theoretically motivated property of models of V1. Also, I encourage the authors to include the analysis on diverse surround neurons, which is now shown in the response to reviewers, in the manuscript.

(Remarks on code availability)

Reviewer #2

(Remarks to the Author)

The revised manuscript is much improved, especially the exposition and figures. I had originally raised a number of concerns about methods, controls, and presentation, most of which were addressed. In the revised manuscript, some of my quantitative concerns remain. However, these concerns will almost certainly not affect the qualitative conclusions of the paper. These quantitative concerns also could be levied against many comparable papers in the literature. So, I leave it to the editor to decide whether these concerns warrant further revisions.

Lingering minor quantitative concerns

1. I originally raised a minor concern having to do with EM vs moment-matching. I think the authors mistook this as a concern about having two separate models vs a single combined mixture model.

Allow me to restate the (minor) concern. The models are first fit to natural images using a moment-matching method to estimate the prior covariances. The shared-modulator and independent-modulator models are then compared (in Fig 2) by comparing their marginal likelihoods in various configurations. Let Σ_{MM} be the moment-matched prior covariance, and let Σ_{MLE} be the prior covariance one would get using MLE. I suspect (but might be wrong?) that Σ_{MLE} and Σ_{MM} are very similar matrices, but not exactly equivalent. My concern was then that $\log[p_{\text{shared}}(x|\Sigma_{\text{MM}})]$ is not exactly equivalent to $\log[p_{\text{shared}}(x|\Sigma_{\text{MLE}})]$, and likewise for $p_{\text{independent}}$. Again, I suspect that the effect on the conclusions of the paper will be nil even if the matrices are distinct.

If it's the case that Σ_{MM} is equivalent to Σ_{MLE} , I'd suggest a minor edit around L198 to clarify that the comparisons in Fig 2 are, in fact, comparisons of *maximum* marginal likelihoods.

2. I originally raised a concern about "Insufficient details in the experimental methods and reporting of differences between subjects." I separately raised a concern that "I am wary of statistical tests on correlations..." Allow me to revisit both of these concerns together. The authors have partially addressed the first concern by including session details in Fig 4B and Table 1, and they have partially addressed the second concern by including Supplemental Fig 19.

- Lingering concern 2a: Table 1 shows that subject a2 had 6 repeated sessions, while the other three subjects each sat for a single session. This raises a statistical concern in how the quantities in and around Fig 4 are calculated: repeated sessions from the same chronically-implanted Utah array do not constitute independent data points, but are "pseudo-replicated" data points. The t-tests that the authors use are only appropriate for independent data. Where the authors report values of $p < 0.001$, it is plausible to me that a corrected estimate might be $p < .05$ or even $p < .01$. The methods as stated suggest inflated significance due to pseudo-replication of a2's data. Again, I don't suspect the qualitative conclusions will change, but I do suspect that the *statistical significance* of the reported changes in correlations are inflated. A simple way to address this would be to report effect sizes and significance separately for each session. This has the reverse effect of under-inflating significance, but this is arguably preferable from a replicability standpoint. Another way to address it would be to choose any one of a2's sessions at random when calculating effect sizes for combined subjects.

- Lingering concern 2b: analogous to the previous concern, doing t-tests on mean correlation values makes a similar error. In my first review, I fear I may have made some misleading comments about how to address this. I appreciate the authors' extensive efforts to fit regularized covariance estimates to their data (L1033-L1036). Unfortunately, this missed my point. The root of my concern remains in Fig 4a and in the p-values reported in L355 and L358. This is again a concern about inflated significance due to using a statistical test that assumes independence where the data are not actually independent. What the low-rank estimator of Yatsenko et al. provides is not just an alternative way to calculate correlations, but a way to approximate the number of degrees of freedom in the analysis. If the (mean-matched) *changes* to the noise covariance structure between small- and large- images is well-described by some rank- k change to the covariance, then the true degrees of freedom are more like $n \cdot k$ for a population of n neurons. The current methods in the manuscript, as I understand them, use a t-test to compare mean correlations by taking all $n(n-1)/2$ off-diagonal elements of the correlation matrix.

A quick-and-dirty correction would be to perform a t-test using $n \cdot k - 1$ degrees of freedom rather than $n(n-1)/2$, using $k \approx 3$ since that's what the authors report in their rebuttal as the lowest rank which gives good fits.

Another simple approach might be to design some sort of data-shuffling technique and use a permutation test, but the authors would need to think carefully about what shuffling method would be appropriate.

Finally, a third alternative would be to reframe the hypothesis entirely as a prediction for changes to the population *variance* along a particular dimension of response-space rather than a prediction for changes to the *mean correlation*. Indeed, "mean correlation" as a population statistic is only indirectly of theoretical interest. The directions and magnitudes of population noise-covariance are more directly relevant to signal detection theory. For this reason, I think that the analysis in Supp Fig 19 is both more statistically honest and more informative from a theory standpoint since it uses a *paired* test for changes in population *variance*.

I'm suggesting any of the above options. Option A is the authors might keep the paper as-is with some rough corrections to the reported p-values for mean correlations or, at the very least, a caveat in the text that the reported p-values are likely inflated. Option B is to do more thorough corrections through careful modeling of the null distribution for low-rank changes to

covariance or using a carefully constructed permutation procedure. Option C is to more substantially edit so that "mean correlations" are not the quantity of interest; essentially, replace the correlation analyses in Fig 4 and beyond with the population variance analyses in Fig 19 (further accounting for pseudo-replication across sessions, etc...). Option D is that the authors come up with their own sensible approach to appropriately "deflate" their statistical tests.

Concerns that were addressed

1. Replicating (sanity-checking) some key marginal statistics of the model: now done in Supp Fig 16. (It would be useful to state that the 'single neuron' analyzed is representative of the population).
2. Validating that models seen in prior work *do not* show the key effects: now done in Supp Figs 14-15. This is done using ablations on the present model, which I think is a nice way to address the original concern while highlighting the innovation here.
3. Missing references are now included, and differences with the present work are nicely explained.
4. My original comment about MAP vs full-inference over modulators is addressed. I was mostly wrong about Simpson's paradox being the key to understanding the positive correlations. But I'm glad that this comment did lead to clearer interpretation of the source(s) of the observed effects in the Discussion and in Fig 18.
5. Figures are overall improved. I appreciate the use of contour plots to show the true posteriors rather than gaussian approximations.

(Remarks on code availability)

Reviewer #3

(Remarks to the Author)

I appreciate the hard work the authors have put in to revising this paper. It is much improved, but it is still quite hard to follow in places. My comments below include one key point and a series of suggestions for greater precision/clarity in phrasing/definitions.

1. The overarching finding of this paper is that the shared vs independent modulators operating on a V1-like circuit can produce different outcomes depending on the spatial scale of the images that the circuit is responding to. This raises the issue of how the brain 'decides' over what spatial scale (in V1) shared vs independent modulators might operate, since it can't know a priori what the visual stimuli are. I raised this point in the previous review (R3.7 "How do you envision the brain knowing ahead of time whether two features are part of the same object and therefore deploying the same modulator?"). The authors have added a section to the Discussion regarding divisive normalization and normalization pools, but I don't think this addresses the comment. Conceptually, I'm interested in the authors' thoughts/speculations about whether they envision the shared/independent modulation scheme is adjusted dynamically based on some initial response to the visual stimuli, or is it hard wired at some scale across V1 representation based on how the circuit has developed in response to typical scene statistics, or something else? Ultimately, my comment is about making sure any critiques about potentially circular reasoning are considered and addressed.

2. Outsiders to the field could still have difficulty understanding the reasoning behind the claims. I'll try to be as specific as I can.

a. Does the term "image" always refer to the visual stimulus, or does it (sometimes?) refer to the postulated inference about what the visual stimulus is based on the neural activity in V1?

b. Does the term "modulator" refer to a postulated neural circuit element, a mathematical construct capturing correlated spatial context intrinsic to natural visual stimuli (i.e. part of the external world, not part of the postulated neural circuitry), or something else?

c. Does the term "feature" refer to aspects of visual stimuli, aspects of V1 receptive fields, or?

d. Introduction, lines 75-78: "here we consider a simple generative model known as Gaussian Scale Mixture (GSM29). The key assumption of this model is that a global 'modulator' variable modulates multiple features and thus introduces statistical dependence among them (details in Fig. 1A and in Results)". What do 'modulator' and 'features' refer to here? Is the modulator a brain signal? Are the features attributes of the image, attributes of neural tuning functions, activity levels of neurons (maybe in multiple neurons), or something else entirely?

e. (Minor) "Past work strongly supports this theory for single neuron activity" – does "this theory" refer to the Gaussian Scale Mixture model or the sampling hypotheses? Both are mentioned in the preceding paragraph.

f. Results, first two paragraphs: "To study the relationship between pairwise neural responses and image statistics"... In the single neuron GSM (Fig. 1A), an image is generated from linear combinations of localized oriented features, each modulated by a Gaussian coefficient (a latent variable g).

Does "image" here refer to a visual stimulus, or to some neural representation of that stimulus? The juxtaposition of 'neural/neuron' with 'image' in these two sentences leaves the meaning unclear.

Without understanding which we are referring to, it becomes impossible to be sure what is meant by 'localized oriented features', or what is being postulated as modulating those features. Are the 'localized oriented features' receptive fields of input neurons? Or does this text solely refer to visual stimuli, and how they were generated for testing of a neural model?

Figure 1's legend didn't provide clarification on these questions, but the Methods did provide some help, and some of these details should be incorporated

These are some examples from early in the paper. If the reader is set up well for understanding early on, it will be worth the effort.

(Remarks on code availability)

Version 2:

Reviewer comments:

Reviewer #1

(Remarks to the Author)

In line with my review in the previous round, my concerns and comments have been addressed, and support the publication of the paper.

(Remarks on code availability)

Reviewer #2

(Remarks to the Author)

The authors have addressed all of my concerns, and I appreciate their time and thoughtfulness towards increasing the precision and rigor of the claims, especially the inclusion of Fig 18, Table 1, and the adjusted hypothesis-testing procedure using subsampled pairs.

(Remarks on code availability)

Reviewer #3

(Remarks to the Author)

The authors have done a fine job responding to my comments. I have no further suggestions.

(Remarks on code availability)

Reviewer 1

Remarks to the Author:

The paper presents a study on stimulus-dependence of noise correlations in V1. The paper introduces a model-based hypothesis on how noise correlations between pairs of neurons is modulated by changing the aperture in which natural stimuli are presented and this hypothesis is tested on multi-site electrophysiological measurements. The key insight of the paper is elegant and is based on a rather fundamental property of probabilistic modelling of visual processing: uncertainty in V1 is determined by multiple sources of variability some of these imply independent variability others carry shared variability; the authors point out that upon changes in stimulus presentation aperture predictable changes occur in the relative strengths of these sources of uncertainty, resulting in changes in measured noise correlations. This insight is both novel and important, and makes a significant contribution to population-level interpretation of the neural code, thus going beyond single-cell-level coding strategies. The experimental recordings are well aligned with the theory and provide a convincing case for the theoretical predictions.

Response: We are grateful for your positive evaluation.

R1.1: I am raising two issues related to the theory/recordings and two additional issues related to the presentation. First, the model consists of two ‘focus’ neurons that are flanked with a population of ‘surround’ neurons. These surround neurons feature similar sensitivities as the focus neuron, their receptive fields differ from the focus neuron only in the location of the RF center. This choice seems to be very specific but I could not find clear motivation for restricting analysis to such a model, and for retaining the generality of the model under the specific choice the authors made. Demonstrating the generality of the predictions of the theory would strengthen the paper.

Response: Thank you for your thoughtful feedback. We have modified the text to motivate our primary choice of surround filters, and conducted extensive new simulations with different filter arrangements to demonstrate the generality of our model’s predictions.

First, to clarify, we develop models with two ‘focus’ neurons so we can study pairwise correlations, and we include ‘surround’ filters to model the well-established surround modulation of the activity of each ‘focus’ neuron. Surround neurons also allow studying how spatial context in images shapes the co-variability between two focus neurons: Without surround neurons, only direct pairwise interactions between the focus neurons would be captured, missing the broader image structure’s influence on joint responses.

Our specific choice to use eight ‘surround’ neurons with similar orientation preferences to the ‘focus’ neuron is based on two main considerations: 1) empirical evidence: V1 neurons exhibit enhanced surround modulation of trial-averaged activity when the orientations of the stimuli inside the receptive field (RF) and in the surround are similarly oriented Walker et al. (1999); Cavanaugh et al. (2002); Hashemi-Nezhad and Lyon (2012). This suggests that models of surround modulation should include some form of orientation tuning of the surround filters. 2) Consistency with previous research: we adopted this arrangement in our previous study Festa et al. (2021) for the same reasons, and found that surround suppression of single-neuron response variability is also similarly tuned, although weakly.

In addition, as suggested by the reviewer, we have explored the effects of varying the orientation preferences of the surround filters relative to the center. Our new analysis revealed varying degrees of modulation, depending on the relative tuning of the filters and the structure of natural images. However, when averaging across multiple natural images, the qualitative predictions of correlations remain consistent. Thus, our model’s predictions are robust and

generalizable beyond the specific configuration initially presented. We have reported this in the manuscript and in the reviewer figure below. We can add the figure to the supplementary materials if the reviewer or editor believes it is required.

Effect of surround filter orientation on correlations modulation. Changing the orientation of surround filters alters the level of correlations modulation for individual natural images (thin lines) due to their distinct structures. However, when the modulation is averaged across multiple natural images, the overall level of modulation remains similar (thick lines and circles). The orientations of the two neurons are specified as ‘orientation 1/orientation 2,’ where the first orientation corresponds to neuron 1 and the second to neuron 2. The filled gray circles correspond to the configuration used in Fig. 3G of the manuscript.

Manuscript changes: (Methods)

The motivation behind choosing common orientations for center and surround filters is twofold. First, several studies Walker et al. (1999); Cavanaugh et al. (2002); Hashemi-Nezhad and Lyon (2012) demonstrated that neurons in V1 exhibit enhanced modulation when stimuli inside the RF and in the surround are similarly oriented, indicating orientation-tuned surround modulation. Second, we adopted this arrangement in our previous single-neuron model to capture the tuning of response variability Festa et al. (2021). Thus, our choice ensures that the model we use here captures known single-neuron surround modulation of both mean spike count and response variability (Supplementary Fig. 16). Nonetheless, because altering the orientation preferences of the surround filters can lead to different degrees of modulation depending on the structure of the visual input, we conducted additional simulations to verify the generality of our results: when averaging across a variety of natural images, the qualitative predictions for how correlations are modulated by stimulus size remain consistent regardless of the tuning of surround filters.

R1.2: Also, it would be beneficial for the readers, what single-cell properties of the Gaussian Scale Mixtures model this simplified version retains.

Response: As explained above, and shown in a new Supplementary Figure 16 (reported also here), our model retains key single-neuron properties identified in previous studies, including the suppression of mean spike count and reduction in response variability (Fano factor) by surround stimuli.

Supplementary Figure 16: Single-neuron properties in a single GSM model neuron. We analyzed a centered neuron’s mean spike count and Fano factor for small and large images. Only the top 20% of images with the highest center filter responses were included, excluding non-responsive images. These results align with previous studies, demonstrating that our model preserves key single-neuron properties observed experimentally.

R1.3: Second, Figure 4 presents the experimental confirmation of the theoretical predictions. As far as I understand, the theoretical analysis assumes that the two focus neurons are both stimulated with different aperture images and it is the natural image statistics that predict disjunct modulators for the two focus neurons that results in the facilitation of noise correlations when focus neurons have distant RFs. In the case of the experiment, however, a slightly different scenario is presented: only one of the neurons is stimulated with the small aperture image, the other neuron has a grey image in its RF. This difference between theory and experiments is not evident in the text. Although the authors perform a large number of controls in which criteria for centered and mixed pairs are exhaustively explored, this specific distinction is not made explicit. I am wondering if a more direct test of the experimental setup can be performed with the model.

Response: The arrangement of the two focus neurons in the model of Figure 3 aligns exactly with the arrangement of experimental neuronal pairs in Figure 4. Therefore, for both experimental and model mixed pairs, the visual input to the RF of the off-centered neuron was uniform gray when we presented small images.

We agree that this was not described sufficiently clearly in the original manuscript, therefore we have revised the text in the model section ‘Pairwise models and image statistics predict when surround stimulation suppresses or facilitates correlations’ (e.g. Line 228), and in the section describing Figure 4 (Line 319). We have also added two schematics in Figure 3A and C to illustrate the overlapping and non-overlapping neuron pairs. These visuals clarify how our model configurations map onto the experimental conditions.

Specifically, in Figure 3, in the shared modulator model the two neurons have overlapping RFs centered on the small image. This coincides with the RF arrangement of the experimental pairs that we classified as ‘centered’ (both neurons’ RFs are centered on the small image). In contrast, the independent modulator model in Figure 3 uses focus neurons with distant (i.e. non-overlapping) RFs: one neuron is centered on the small image and the other is off-centered. This corresponds to the experimental ‘mixed’ pairs.

Comments on writing:

R1.4: First, the paper introduces the Gaussian Scale Mixtures model as early as the introduction as a model of natural images. Then, the results introduces a very constrained version of the

Gaussian Scale Mixtures model, that is not really a model of natural images but shares some characteristic features with the original model. I believe that a more careful transition to the model actually studied in the paper would be utterly beneficial. Without a proper motivation, the toy-looking model seems unconvincing as a model of natural images and the reader gets lost on the assumptions and approximations behind the model. This is especially important, as the discussion states that the theory is parameter-free as its parameters are fully determined by natural image statistics. While the statement is true, the model is also largely determined by the introduced simplifications, which could limit its generality and the reader needs guidance with respect to its generality.

Response: To address this point, we have revised the Introduction (Line 75) to include a more intuitive explanation of the model studied in the paper, providing a clearer transition from the Gaussian Scale Mixture model of natural images to the version analyzed in our work. The revised introduction (Line 98) also places more emphasis on the key conceptual advance of modeling both shared and independent sources of uncertainty about image features. Additionally, we revisited the Results section to ensure consistency with the revised Introduction.

R1.5: Second, the paper makes important and exciting contributions to understanding pairwise statistics of neural responses. To motivate the approach, the introduction explores earlier contributions. While the paper rightfully highlights that traditional approaches focus on single neuron responses, several works started addressed stimulus-dependence of noise correlations, which is central to interpret noise correlations as important components of coding. These studies include Lin (2015 Neuron, Franke (2016) Neuron, Banyai (2019) PNAS, Csikor (2024) biorxiv. In fact, latent variables that jointly modulate V1 responses and contribute to noise correlations has a wider literature, that spans papers such as Haefner (2016) Neuron, Lange & Haefner (2017) Curr Opin Neurobiol, Bondy (2018) Nat Neuro, Banyai (2019) Curr Opin Neurobiol.

Response: Thank you for highlighting these prior studies which we had not fully acknowledged in the original submission. We have included the references in the Introduction and we have modified Discussion (“Modulation of correlations by other stimulus features and attention”) to discuss the references most closely related to our work.

Manuscript changes: (Discussion)

Another study conceptually related to ours invoked probabilistic inference to understand pairwise V1 response statistics to synthetic textures and natural images Bányai et al. (2019). They showed that patterns of correlations depend on high-level statistics, more than on low-level statistics. They explained the finding as reflecting that probabilistic inferences about high-level features (computed by higher visual cortex and fed back to V1) set the context for the inferences in V1. Different from our work, Bányai et al. (2019) did not model natural image statistics or pairwise neural responses explicitly, and therefore did not make detailed predictions about what aspects of images enhance or suppress correlations. Trainable models for hierarchical inference Csikor et al. (2023) that reproduce the data of Bányai et al. (2019) could be applied to the stimulus manipulations we have considered. More broadly, other studies also support the view that perceptual inferences about task-relevant latent variables jointly modulate V1 responses, including correlations, via feedback Haefner et al. (2016); Lange and Haefner (2017); Bondy et al. (2018); Bányai and Orbán (2019); Roelfsema (2023). Therefore, in addition to the recurrent mechanisms we discussed above, top-down feedback could contribute to the diverse modulation of V1 correlations by image size.

Minor:

- line 109: Bilal Haider’s paper on the surround stimulation on reliability and sparseness seems especially relevant at this point

Response: Thank you for pointing this out. We have added this study as a reference for surround stimulation.

- line 147: the notation is not introduced with sufficient care: the reader might easily get lost what it means

Response: We have added details to improve the clarity of the notations (Line 141).

Manuscript changes:

Our primary objective is to estimate the joint posterior probability distribution of the features encoded by two neurons, given an image, denoted as $p(g_{c1}, g_{c2}|x)$. The numbers indicate the model neuron, and c denotes centered features (g_{c1} : pink, g_{c2} : beige in Fig. 1B, left).

- Fig 1: the step from posterior to sampling is indicated as inference, while the text refers to this step as representation of inference

Response: Indeed! Corrected.

- lines 164–169: this statement seems very dense, I believe that some extra explanation would be welcome for the general reader

Response: We have expanded these lines (Line 155-185) to provide a more detailed and accessible explanation for the general reader.

Manuscript changes: (Results)

The use of these two distinct GSMs was motivated by observations on the statistical properties of natural images, which often exhibit non-stationarities: namely, statistical dependencies can vary across regions. Due to these non-stationarities, features within the same visual object tend to be statistically dependent, influenced by common underlying factors. These dependencies are effectively captured by a GSM model with a single global modulator that scales all features of an object. In contrast, features belonging to different visual objects are generally more independent, as they are influenced by separate factors Coen-Cagli et al. (2009). For such cases, using independent modulators that scale each feature separately, better captures the statistical independence of those features.

For the pairwise application considered here, we reasoned that joint inferences about pairs of features (represented by pairs of neurons) should account for whether these features are *a priori* more likely to be part of the same or different visual objects. This prior probability depends on factors such as similarity (e.g., orientation preference) and spatial proximity. Intuitively, features that are similar and located close together are more likely to belong to the same object and are thus better modeled with a shared global modulator. Conversely, dissimilar or distant features are more likely to belong to different objects, making independent modulators more suitable. By incorporating this reasoning into our model, we aim to capture the statistical dependencies (or lack thereof) between different features in natural images. Next, we tested this intuition formally.

- lines 181–183: training remains elusive here: the reader would benefit from some explanation what aspects of the model are trained

Response: In response, we have expanded these lines (now Line 196) to provide a more detailed explanation of the aspects of the model that are trained. Specifically, we now describe how the covariance matrix between filter outputs is computed using moment matching on natural images.

Manuscript changes:

To study how well the shared and independent GSM capture image statistics, we computed the likelihood of each natural image under each model, and compared the models by their log-likelihood ratio. We implemented the GSM for each neuron similarly to past work on surround modulation, i.e.

with a group of bandpass linear filters covering a reference location and eight surrounding locations (Fig. 2A; details in Methods). The model parameters, i.e. the prior covariance matrices of the local latent features, were estimated with moment matching Doulgeris and Eltoft (2009) from an ensemble of 10,000 natural images from the ImageNet validation set Russakovsky et al. (2015).

- Fig 3a caption does not specify the top and bottom subpanels.

Response: We added additional panel labels to clearly distinguish between the top and bottom subpanels and specify their content.

- line 343: The mean matching procedure seems to be central to the findings. As far as the short description permits, the procedure is identical to the one described in Banyai (2019) PNAS. Based on Fig S12, the changes in noise correlations both for central and mixed pairs are similar to those that would be expected from de la Rocha (2007) therefore a direct comparison of the effect of aperture change *with* and *without* rate matching could be instructive. I applaud showing rate matched results but a convincing demonstration of the distinct effect of the theory and mere firing rate changes are important.

Response: Thank you for highlighting this point. To address this, we have performed new analyses and added a new supplementary figure (Supplementary Fig. 17) that compares correlations with and without mean-matching of spiking activity. Our mean-matching procedure is similar to that described in Banyai et al. (2019), where we matched between conditions the distributions of average spike counts by subsampling the data to ensure that differences in mean firing rates do not confound any differences in correlations. The results show that the modulation of correlations with aperture size is qualitatively similar with or without mean-matching.

Supplementary Figure 17: Effect of mean-matching on correlations estimate. Spiking activity can directly influence estimates of correlations (De La Rocha et al., 2007; Cohen and Kohn, 2011; Schulz et al., 2015). To eliminate the potential confounding effect of changes in spiking activity due to image size on the correlation measurements, we performed a mean-matching analysis. Specifically, we examined how correlations are modulated by aperture size, both with and without mean-matching of spiking activity. In brief, for each pair of neurons, we calculated the average spike count across trials for each image and size condition. We then constructed histograms of these neural-pair-averaged mean responses separately for small and large images. To create mean-matched histograms, we subsampled the data by selecting the minimum number of samples per bin across conditions, ensuring that the distributions of average spike counts were matched between small and large images (Top panels, gray bars; see Methods for details). From these mean-matched samples, we computed the corresponding correlations. For comparison, we also randomly sampled from the raw distributions using the same number of samples as in the mean-matched condition. The results show that the modulation of correlations with aperture size is preserved both with and without mean-matching (two-sided t-test against the null hypothesis of no difference: p -value < 0.001 ; n.s. (not significant)), which indicates that changes in spiking activity due to image size do not account for the observed effects on correlations. Error bars represent standard error. Data from all sessions are aggregated.

- Refs Bawat et al, Schrimpf et al seem to be corrupted.

Response: Thank you. We have reviewed and corrected the references to Rawat et al. and Schrimpf et al., ensuring that all citations are accurate and properly formatted.

- throughout the equations (eg lines 679 & 720): subscripts and superscripts ('shared') are not placed in mathrm .

Response: Thank you for pointing this out. The labels shared and independent are now typeset in an upright font format for clarity.

Reviewer 2

Remarks to the Author:

At a high level, this paper (1) makes a novel observation about a qualitative phenomenon in an existing model family (the GSM), (2) formulates this as a (qualitative) prediction for pairwise correlations among neurons in V1 using an established framework (neural sampling), and (3) finds evidence for the predicted trends in new data collected from anesthetized macaque V1. Building bridges between theory and data in this way is challenging and important, and I think that the work here is high quality.

Response: We are grateful for your thorough reading and positive evaluation of our study and its main findings.

However, I feel that the novelty of the work is overstated (some key references missing). It is also an unfortunate standard in this field that novel predictions are so often evaluated on novel data without (1) benchmarking the predictions on extant data, nor (2) benchmarking earlier models on the new data. Still, I think this paper contributes some genuinely novel and interesting insights on the relation between sampling (in a GSM) and noise correlations.

Response: As detailed below, first we acknowledge that some key references were initially missing (noted also by R1) and have now included them. In doing so, we have expanded the Discussion to clarify the innovation of our work relative to those prior studies. Second, we have modified the text and added new supplementary figures to clarify that our new model reproduces extant data captured by earlier models (Supplementary Fig. 16), and that the earlier models cannot trivially capture the new data (Supplementary Figs. 14-15). We agree that thorough benchmarking of this kind would benefit the field, and this was a key consideration in developing the modeling work for this paper. Please see detailed responses below.

R2.1: The authors state in the introduction that predictions from the neural sampling hypothesis have been limited to marginal statistics (individual neurons' means and variances) without addressing pairwise statistics (covariances). This is not true; see, for instance, refs [1-4] below (none of which is currently cited).

Response: Thank you for bringing these references to our attention. You are correct that predictions from the neural sampling hypothesis have addressed pairwise statistics, as demonstrated by the studies you mentioned. We now cite these references in the revised Introduction and Discussion. We have also clarified in the Discussion ("Modulation of correlations by other stimulus features and attention") how our work provides substantial new insight into neural encoding, relative to these prior studies. Briefly, the main advance is that we model explicitly statistical dependencies across space in natural images, and in so doing we arrive at a detailed understanding of the factors that determine surround suppression versus facilitation of V1 correlations.

Manuscript changes: (Discussion)

Another study conceptually related to ours invoked probabilistic inference to understand pairwise V1 response statistics to synthetic textures and natural images Bányai et al. (2019). They showed that patterns of correlations depend on high-level statistics, more than on low-level statistics. They explained the finding as reflecting that probabilistic inferences about high-level features (computed by higher visual cortex and fed back to V1) set the context for the inferences in V1. Different from our work, Bányai et al. (2019) did not model natural image statistics or pairwise neural responses explicitly, and therefore did not make detailed predictions about what aspects of images enhance or suppress correlations. Trainable models for hierarchical inference Csikor et al. (2023) that reproduce the data of Bányai et al. (2019) could be applied to the stimulus manipulations we have considered.

More broadly, other studies also support the view that perceptual inferences about task-relevant latent variables jointly modulate V1 responses, including correlations, via feedback Haefner et al. (2016); Lange and Haefner (2017); Bondy et al. (2018); Bányai and Orbán (2019); Roelfsema (2023). Therefore, in addition to the recurrent mechanisms we discussed above, top-down feedback could contribute to the diverse modulation of V1 correlations by image size.

R2.2: Regarding model benchmarks, the core of this paper is a prediction about stimulus-dependent correlations in a variant of the well-studied GSM model. The authors note that other papers with other model variants in the GSM family have qualitatively reproduced a number of marginal statistics of V1 activity (e.g. contrast-dependence, tuning curves, variance and fano factor, etc.). Do all of those same marginal statistics still hold in the new model? I would argue that such sanity-checks are a necessary bar to clear before addressing novel predictions of the new model. (I don't expect this is true here, but one can in-principle imagine constructing a model that makes new and interesting predictions for pairwise neural statistics but fails to recapitulate basic facts about their marginal statistics).

Put more succinctly: for a new model M2 to supersede an earlier model M1, M2 should explain everything that M1 explains, and more. It is an unfortunate standard in our field that new models are so often tested on new data without replication or generalization tests across models/datasets.

Response: We appreciate this point. Our model is indeed built upon the single-neuron GSM framework, known to capture a range of marginal statistics of V1 activity, including mean responses and variances, as clarified in the revised text. Further, to demonstrate these marginal statistics remain valid in our extended model, we conducted specific sanity checks by examining individual neurons within the pairwise model configuration. Specifically, we confirmed that single neuron responses from our pairwise model retain the fundamental GSM model properties. As shown in new Supplementary Fig. 16, the modulation of mean spike count and Fano factor match those observed in previous studies and match the predictions of the single-neuron GSM model of Festa et al. (2021).

We also agree that it is important to show that the old models cannot explain the new data. That the single-neuron GSM of past publications cannot reproduce correlations would be trivial if we simply considered two such neurons in isolation (i.e. no correlations). Instead, we considered extending the classical single-neuron GSM to a pairwise GSM either with globally shared modulator or independent modulators. The key innovation of our new modeling is that the choice of shared or independent GSM is dependent on the RF properties of the two neurons. Therefore, the comparison is against a pairwise model that uses shared GSM for all pairs (Supplementary Figure 14; a globally shared mixer is the typical choice in other published papers relating GSM to correlations) and against a pairwise model that uses independent GSM for all pairs (Supplementary Figure 15). Those figures show that neither model can capture our new experimental observations on correlations.

Manuscript changes: Methods (Line 736)

[...] Thus, our choice ensures that the model we use here captures known single-neuron surround modulation of both mean spike count and response variability (Supplementary Fig. 16). [...]

Supplementary Figure 16: Single-neuron properties in a single GSM model neuron. We analyzed a centered neuron’s mean spike count and Fano factor for small and large images. Only the top 20% of images with the highest center filter responses were included, excluding non-responsive images. These results align with previous studies, demonstrating that our model preserves key single-neuron properties observed experimentally.

R2.3: Notably, the model by Orban et al (2016) was also a GSM but they used MAP inference over the modulator v . The present paper replaces this with joint inference over (v, g) . If my understanding is correct (see comment on Simpson’s paradox in Minor point 1 below’), then this simple modification of replacing MAP inference with full inference may be *the* key driver of the predictions about correlations. The authors may want to illustrate this explicitly, showing that MAP inference of v does *not* make the same predictions, which would be a major point in favor of the present work above and beyond the (otherwise very closely related) prior work by Orban et al.

Response: Thank you for raising this excellent point which motivated us to compare more systematically our simulations with the MAP approximation. However, first we would like to clarify that the key innovation of our work relative to Orban et al 2016, is the insight that the GSM modulator can be either shared or independent across neurons. This is dictated by natural image statistics, leading to the predicted opposite effects of surround modulation that we confirmed experimentally. This is now clarified in the Discussion.

Per your suggestion, we have also conducted extensive new simulations and added a paragraph in the Discussion (and a related Supplementary Fig. 18). Specifically, we compare three scenarios reported in the figure: (1) marginalizing the global modulator v by sampling the joint posterior, (2) marginalizing approximately by using the MAP estimate of v , and (3) fixing v at a constant value (i.e. no marginalization). The main finding is that only when we include marginalization, either exact or via MAP, do we reproduce both suppression of r_{sc} with shared modulators, and facilitation of r_{sc} with independent modulators. Furthermore, consistent with your intuition, the MAP approximation fails to account for the variability introduced by the full posterior distribution of v , and so it leads to overall weaker correlations in the shared GSM and stronger correlations in the independent GSM.

Supplementary Figure 18: Comparing marginalization, maximum a posteriori (MAP) estimation, and fixed global modulator values. This analysis contrasts three approaches for handling the global scaling variable: (1) marginalizing it, (2) using the MAP estimate, and (3) fixing it at a constant (image-independent) value. The top row of the figure presents results for the shared modulator model, while the bottom row illustrates the independent modulator model. In panels (A-B) and (E-F), the left panels show, for a single image, samples from the joint posterior of g_1 and v_1 (top) and g_2 and v_2 (bottom). In the shared model, v_1 and v_2 are identical due to the single global modulator. A key observation (right panels) is the noticeable difference in correlation when comparing samples near the MAP estimate of v (top) to those obtained by marginalizing v (labeled "all"; bottom). Panels (C) and (G) systematically vary interval sizes around the MAP estimate to measure correlations across corresponding samples, for multiple images driving the centered filter. While changes in interval size affect correlations as predicted by the effects of marginalization, the qualitative differences between small and large intervals remain consistent. To investigate further, panels (D) and (H) replace interval-based sampling with a direct comparison of posterior distributions. Specifically, we first capture the posterior distribution of the global modulator for each test image, then compare results obtained by replacing marginalization over the global modulator with its MAP estimate. We also test a condition where the global modulator is fixed at a value of 1. Results indicate that MAP estimation approximates marginalization, capturing the overall trend despite being a simplified representation of the full distribution. This occurs because MAP estimates differ in magnitude between small and large images, leading to suppression or facilitation of correlations, depending on whether the global modulator induces shared or independent variability. Conversely, fixing the global modulator at a constant value eliminates differences in correlation between small and large images, as expected. In summary, while the MAP estimate can recover opposite contextual effects of correlation, the magnitude of correlations further depends on marginalization over the global modulator, highlighting its role in accurately capturing variability.

Manuscript changes: (Discussion)

Prior work with GSMs showed that single-neuron V1 activity reflects probabilistic inferences about the image feature encoded by the neuron Orbán et al. (2016); Echeveste et al. (2020); Festa et al. (2021); Aitchison and Lengyel (2016); Goris et al. (2024) and also captured some aspects of interactions between V1 neurons Orbán et al. (2016); Aitchison and Lengyel (2016); Echeveste et al. (2020). **Our study goes substantially beyond that past work, through a detailed analysis of natural image statistics with surprising implications for V1 covariability.**

By building explicit pairwise GSM models, we showed that simply extending the GSM to pairs of model neurons is not sufficient to capture the statistics of natural images. **This is because the assumption made in those prior studies, that a shared global modulator variable scales the local**

features encoded by all neurons, breaks down when those features are sufficiently distant or different (Fig. 2). This implies that a GSM with independent modulators is a better generative model for neuron pairs encoding distant or different features. [...]

This observation about image statistics led to the key new insight of this paper: because increasing image size reduces uncertainty due to the modulator, and thus reduces response variability in sampling-based representations Orbán et al. (2016); Festa et al. (2021); Goris et al. (2024), our pairwise models predicted opposite effects on covariability depending on whether the modulator is shared or not between neurons (Fig. 3). Previous V1 models most closely related to ours Orbán et al. (2016); Aitchison and Lengyel (2016); Echeveste et al. (2020), may only capture the suppression of correlations by spatial context, not the facilitation, because those models assumed shared modulators only. We confirmed that these results require that the probabilistic inference about image features takes into account the modulators in the GSM (i.e. marginalization, see Methods; Supplementary Fig. 18).

R2.4: Insufficient details in the experimental methods and reporting of differences between subjects. It appears in Fig 4b that effects for the utah-array recordings and neuropixels recordings may be qualitatively different. It is unclear from Fig 4b which sessions correspond to which subject. It would be helpful to see more detail on individual subjects’ recording sites, number of isolated units, putative differences in cell types or cortical layer, etc. This is especially crucial in any analysis of correlations, since correlation estimates can be contaminated by mislabeled spikes (in the neuropixels data) and by background fluctuations in the utah-array data [5].

Response: We have added a table summarizing the subjects, number of isolated neurons, and cortical layers. All recordings were conducted in V1, and information on cell types is not available for the experiments. To address potential variability in neuron counts due to distance thresholds and responsivity, we have included results for different thresholds in Supplementary Fig. 10. Additionally, we have updated the session names with a new naming convention for clarity.

Quantitative differences between Utah array and Neuropixels recordings may stem from factors such as sampling bias, electrode characteristics, experimental conditions, or spike detection sensitivity. Despite these differences, the suppression of correlations for centered pairs and facilitation for mixed pairs is consistent between both recording techniques.

Table 1: Details of the nine recording sessions across four animals.

animal	session	recording type	centered neuron	off-centered neuron	cortical layer
a ₁	s ₁	Utah Array	25	32	L2/3 or 4B
a ₂	s ₁	Utah Array	45	52	L2/3 or 4B
a ₂	s ₂	Utah Array	39	53	L2/3 or 4B
a ₂	s ₃	Utah Array	40	26	L2/3 or 4B
a ₂	s ₄	Utah Array	49	41	L2/3 or 4B
a ₂	s ₅	Utah Array	22	52	L2/3 or 4B
a ₂	s ₆	Utah Array	50	49	L2/3 or 4B
a ₃	s ₁	Utah Array	42	61	L2/3 or 4B
a ₄	s ₁	Neuropixels	53	190	All layers

Statistical concerns in data analysis.

R2.5: Why was the mean-matching control necessary in the neural data (L938) if it was not necessary in the model (L780)? It’s of course expected that mean firing rate will impact variance,

but I do not follow the logic that it will impact the correlation estimates, unless the concern is about using a biased estimator of correlation, which could be bias-corrected a number of ways (and would also impact the model predictions)

Response: Thank you for raising this point, which we now clarify in the new text and further illustrate in this letter with mean-matching analysis for the model. The estimation of correlations from spiking activity is known to be influenced by firing rates, as shown in prior studies (De La Rocha et al., 2007; Cohen and Kohn, 2011; Schulz et al., 2015). To account for the potential confounding effects of changes in firing rates due to image size, we followed common practice and performed a mean-matching analysis in the neural data to ensure that variations in spiking activity did not bias the correlations measurements Bányai and Orbán (2019).

We have also applied the mean-matching control to the shared and independent GSM models, and the results (reported in the figure below) were consistent with expectations, confirming that the control does not alter the model predictions.

Mean-matching control applied to the shared and independent GSM models preserves the observed trends of correlation suppression and facilitation.

R2.6: The authors appear to use correlation values computed from the raw (sample) covariance matrices, which will in general give biased estimates of the true correlation [6]. The authors might be interested in using a regularized covariance estimator [7] without the mean-matching procedure. In particular, [7] includes a diagonal + rank-1 covariance estimator that seems appropriate in this setting where the hypothesis on "mean correlation" changes is essentially equivalent to hypothesizing a change in variance along the all-ones vector, i.e. a change in the variance of the population mean, without a commensurate change in other dimensions of the space.

Response: To address this point we conducted additional analyses using the regularized covariance method you recommended and modified the Methods text. In the new analysis, we employed a diagonal plus low-rank covariance model Yatsenko et al. (2015) with ranks 1, 3, and 5 (details in the reviewer's figure below). With one factor, the estimated correlations were lower than those from the raw covariance matrix, but the key contextual effects of correlation were preserved, indicating robustness in our findings. As we increased the number of factors to three and five, the correlations became more similar to those obtained from the raw data, indicating that incorporating more factors improved the model's ability to capture pairwise correlation complexities.

Having determined that the results obtained with regularized covariance estimation are consistent with those obtained using the sample estimate and mean matching controls, we believe that using the latter in our manuscript is the better choice. Our primary focus is on contextual modulation of pairwise correlations and the sample covariance method aligns with standard practices in similar studies (including the closely related Bányai and Orbán (2019)), allowing for straightforward comparison. In contrast, regularized covariance estimation and factor analysis are necessary for characterizing shared variability at the population level, which is the focus of our ongoing work but beyond the scope of this manuscript.

Manuscript changes: (Methods Line 1033)

We also confirmed that our qualitative results on surround modulation of correlations are unchanged when using a regularized covariance estimator (Yatsenko et al. (2015)) instead of mean-matching.

Regularized covariance matrix captures contextual modulation of correlations. We followed these steps: First, we performed factor analysis on the data with 1, 3, and 5 factors to model the covariance structure while controlling for shared variability. For each image and factor, we estimated the regularized covariance matrix using $\Sigma = \Lambda\Lambda^T + \Psi$, where Λ represents shared factors and Ψ captures unique variances. Pairwise correlations were then calculated based on these estimates. Neuron pairs were selected according to distance and responsivity criteria to ensure reliable comparisons (see Methods: Characterization of neuronal responses and inclusion criteria).

R2.7: I am wary of statistical tests on correlation that lump all $N(N - 1)/2$ unique correlation values into a test as if they are independent samples from a population. If, say, the correlation effects are driven by a small set of latent modulators (as in the GSM model), then the number of degrees of freedom in the correlation matrices is on the order of N rather than N^2 . I suspect that this effect is inflating the reported significance of the results.

Given that the stimuli are presented in pairs, where the same image is shown small/large, this suggests to me that there is a stronger statistical test available: following the logic above, the "change in average r_{sc} " prediction is essentially a prediction for the change in variance of the population average activity, and no-change or less-increase in orthogonal directions. Could this be tested within-stimulus by comparing responses to small vs large versions of the same image?

Response: Thank you for your thoughtful comments regarding the statistical tests on correlations. In line with your suggestion, we have implemented a new statistical test comparing responses to small and large versions of the same image. However, this required analyzing the variance and covariance of population-average activity, as follows. For each trial, we calculated the average activity of centered neurons and, separately, off-centered neurons, producing two population-averaged responses. We then analyzed the across-trial variance of the centered population activity and the covariance between the centered and off-centered population activities. Our analysis showed that changes in correlations (our original analysis) align with changes in the variance and covariance of population-average activity (we note that the effects would largely cancel out if population average was computed by pooling all centered and

off-centered neurons together). This analysis has been included as a supplementary figure, along with additions to the Results section. As explained in the previous response, we believe that this approach will be particularly useful for understanding population-level structure, in follow-up work.

Supplementary Figure 19: Variance and covariance of population-average activity consistent with contextual modulation of correlations. For each trial, we calculated the average responses of responsive neurons separately for centered neurons and for off-centered neurons, yielding two vectors with length equal to the number of trials, representing the population-averaged activity of centered and off-centered neurons. From these, we estimated a 2×2 covariance matrix across trials. For centered neurons, we compared the variances of their population-average activity between small and large images. For the mixed group (centered and off-centered neurons), we analyzed the covariance between their population-average activities across image sizes. Notably, the variances of centered and off-centered neurons change in opposite directions, hence the effects would largely cancel out if population averages were computed by pooling all centered and off-centered neurons together. By focusing on the interaction between the groups via covariance, we avoided this bias. To assess the significance of these differences, we performed Levene’s test for variance differences in the centered group and a z-test for covariance differences in the mixed group. Each circle represents one image pair from a recording session. The results show that changes in correlations align with changes in the variance and covariance of population-average activity (centered: 2129 cases, 466 significant, 98% of significant cases showed suppression; mixed: 1056 cases, 240 significant, 99% of significant cases showed facilitation), effectively predicting the contextual modulation effects observed.

Manuscript changes: (Results Line 363)

[...] Lastly, we confirmed that the suppression and facilitation effects were statistically significant on an image-by-image basis (Supplementary Fig. 19).

Moderate/Minor comments

- While the figures are generally very well done, I am concerned that the illustrations of correlated variability in Figs 1C and 3A are hiding a critical detail. The short version: the figures, as drawn, make the posteriors appear Gaussian, but their non-Gaussianity (due to uncertainty about v) is a critical aspect of the theory. These sketches would be clearer if they showed contours of the true posterior.

Response: We have modified the figures 1C and 3A-C as suggested.

- The longer version: As I understand the predictions of the model, the predicted positive correlations among overlapping and similarly-tuned neurons are a case of Simpson’s paradox. That is, for any given value of the modulator v , there is a negative correlation among the g_s (explaining-away); but, uncertainty over v drives coordinated changes in the posterior means of the g_s , resulting in a net positive correlation. (This could be done analytically by decomposing the posterior covariance into the average covariance conditioned on each v , and the covariance of the posterior means across different v s). If this intuition is correct, it would be much clearer to illustrate the Simpson’s paradox effect in the figures.

Response: Based on the new analysis conducted in response to your suggestion (point R2.3), we found that for

a fixed value of the modulator, correlations do not vary between small and large images. However, correlations remain positive among overlapping and similarly-tuned neurons, with their magnitude depending on the difference in orientation preferences between the two model neurons. Please refer to **R2.3** and related figure for more detail.

- I am not sure how to think about the model constraint that all surround neurons have identical but translated projective fields. Would the same predictions not hold for a GSM in which the projective fields are also fit to images? Again in the spirit of model-benchmarking, I would have appreciated at least some discussion on what limitations (or not) there are due to this constraint.

Response: Thank you for pointing this out. We have addressed this in the revised text below and reported new simulation results and further clarification in **R1.1**.

Manuscript changes: (Methods Line 747)

We assumed that all surround neurons have identical, translated receptive fields. While fitting the receptive fields to images would not substantially alter our predictions—since the GSM framework captures the key statistical dependencies in natural images—we acknowledge that this constraint may limit the diversity of receptive field properties represented and thus the quantitative match to the data.

- I was surprised to see that the data were collected from anesthetized monkeys. I suggest the authors at least comment on whether/how they expect the results to change in awake subjects, especially given that anesthesia affects inter-area communication, and (as far as I'm aware) some center/surround effects are thought to be driven by feedback.

Response: We agree that anesthesia might influence feedback signals, which are involved in mediating some center-surround effects. However, studies have shown that surround effects (mean, single-neuron responses) are largely indistinguishable between awake and anesthetized animals. Additionally, most of our current understanding of surround modulation originates from experiments conducted on anesthetized animals. Moreover, using anesthetized animals minimized eye movements and ensure more stable retinal input across trials, reducing stimulus-induced variability.

Manuscript changes: (Discussion Line 664)

Our experimental data were collected from anesthetized monkeys, to minimize eye movements and ensure more stable retinal input across trials, reducing stimulus-induced variability. It is possible that the effects reported here might differ in awake animals, where attentional fluctuations and feedback may influence correlations and center-surround modulation. We note however that surround modulation of single-neuron response mean and variability with natural images is qualitatively similar and consistent with GSM predictions in both awake fixating and anesthetized animals Festa et al. (2021).

- The model-fitting procedures seem non-standard. Why not use the EM algorithm for fitting the GSM (derived by the authors in 2009 work)? What is done here – using empirical covariances of filter responses to natural and noise stimuli – is perhaps an approximation to the MLE parameters (and therefore a good way to initialize EM), but not equivalent to the MLE parameters. It is also not obvious to me whether the extent of this approximation is worse in the shared vs independent models. Ultimately, this complicates the interpretation of the likelihood comparisons in Fig 2 (though it is unlikely to affect the qualitative trends).

Response: We agree that the EM algorithm would be an ideal approach to fit model parameters if we used a probabilistic mixture modeling approach, as in our previous study (Coen-Cagli et al., 2009). There, we developed a mixture of GSM (MGSM) framework, and used the EM algorithm to estimate different covariance matrices for each mixture component. However, this manuscript does not employ an MGSM framework. Instead, we learn two

separate prior covariance structures depending on whether interactions between pairs of neurons are included: the shared GSM assumes interactions, while the independent GSM does not, for all images. With this generative model, similar to Festa et al. (2021), we used an empirical covariance matrix estimated through moment-matching.

We have explained in Discussion (Line 541) that using probabilistic mixtures is an important direction for future work to make image-by-image predictions.

References

1. Bányai, M. et al. Stimulus complexity shapes response correlations in primary visual cortex. *Proceedings of the National Academy of Sciences* 116, 2723–2732 (2019).
2. Bányai, M. & Orbán, G. Noise correlations and perceptual inference. *Current Opinion in Neurobiology* 58, 209–217 (2019).
3. Haefner, R. M., Berkes, P. & Fiser, J. Perceptual decision-making as probabilistic inference by neural sampling. *arXiv* (2014).
4. Lange, R. D. & Haefner, R. M. Task-induced neural covariability as a signature of approximate Bayesian learning and inference. *PLOS Computational Biology* 18, (2022).
5. Schulz, D. P. A., Sahani, M. & Carandini, M. Five key factors determining pairwise correlations in visual cortex. *Journal of Neurophysiology* 114, 1022–1033 (2015).
6. <https://stats.stackexchange.com/questions/148346/shrunken-r-vs-unbiased-r-estimators-of-rho>
7. Yatsenko, D. et al. Improved Estimation and Interpretation of Correlations in Neural Circuits. *PLoS Computational Biology* 11, 1–28 (2015).

Response: Thank you for providing these additional references. As explained above, they are cited and discussed in the new manuscript.

Reviewer 3

Remarks to the Author:

The study by Farzmahdi et al contains important findings and insights regarding the nature of shared vs independent sources of variation and how such sources will affect the representation of natural images in V1. Overall, I like this study and I think that the paper has the potential to have a broad readership and impact. Toward that end, I have suggestions about how best to frame the work so that its importance becomes clear.

Response: Thank you for your positive evaluation of our study. We appreciate your suggestions on how to frame our work to highlight its significance.

R3.1: The first part of the paper describes in detail the GSM model, which the authors are known for developing. It is a useful model on its own terms, but I would respectfully suggest focusing specifically on the design feature involving the shared vs independent modulator parameter “ v ”. The details of the GSM except for this modulator parameter could be couched in very general terms and/or moved to the methods section, which would permit focusing on the central question of the global modulator vs separate modulator signals. Terminology in figures could be “shared modulator”, “independent modulator” etc, to focus the reader on that part of the simulations rather than on the GSM as a whole.

Response: Thank you for your suggestion. In the revised manuscript, we use the terms “shared modulator” and “independent modulator” in the figures and text wherever they help to emphasize the simulations. In the revised Introduction, we have also reduced and simplified the explanation of GSM (Line 75) to place more emphasis on the shared and independent sources of uncertainty (Line 98).

R3.2: Doing so would allow the authors space to present the idea conceptually, which would (presumably) help alleviate one of my confusions – if the modulator is the only source of shared (or not) variability that is provided to the neurons in the modeling section, then how is it the case that there are any noise correlations at all in the independent modulator model (Figure 3A, B – bottom panels)? This led me to wonder if the analysis was mixing signal and noise correlations? Signal correlations get defined and used later in the paper so it seems like that’s not the case but then I’m left without an explanation for this finding.

Response: Thank you for pointing this out. The confusion seems to originate from the assumption that the global modulator is the only source of variability affecting correlations in our models. However, as outlined in the results section and illustrated in the schematic in Figure 3C, there are two sources of uncertainty about the latent variables: variability due to the global modulators (v) and variability due to input noise (η). In the model with independent modulators, correlations arise from the input noise, and can only be reduced by independent variability due to the unshared modulators. This explains the presence of correlations in Figure 3B, despite the independent nature of the global modulators. Importantly, the analysis does not mix tuning similarity and correlations, as tuning similarity is explicitly defined and addressed later in the paper. To clarify this distinction and address potential misunderstandings, we have made adjustments to the text to emphasize the two sources of variability and explicitly explain their roles in the independent model.

Manuscript changes: (Results)

We next asked if the covariability between neuron pairs differs for small and large images, focusing on correlations (often referred to as spike count correlations, noise correlations, or r_{sc} , which measure

the Pearson correlation of spike count responses across repeated identical stimuli) as is commonly done to measure changes in covariability beyond those due to changes in single-neuron variance. [...]

The opposite contextual modulation effects in the models stem from the different sources of uncertainty about the latent features g_{c1} and g_{c2} . This uncertainty is determined by both the global modulators and additive noise, as depicted in Fig. 3E-F. In the shared model, uncertainty about the global modulator is the main source of shared variability among neurons. In contrast, in the independent model, distinct global modulators induce private variability for each neuron. In both models, the input noise (depicted in the brown sections of Fig. 3E-F) contributes to shared variability simply reflecting overlap between filters (see Methods). Importantly, the contribution of the additive noise to variability is not affected by stimulus size (Supplementary Fig. 5). As stimulus size increases, the shared model exhibits reduced uncertainty linked to the shared modulator, thereby decreasing shared variability and, consequently, correlations. Conversely, the independent model typically shows a decrease in independent variability, thereby allowing the other source of correlations (i.e. the additive noise) to become more evident. The schematics in Fig. 3E-F illustrate how changing the image size affects uncertainty regarding the latent features.

R3.3: Generally, both the figures and the text could use some attention at both the flow and detail levels. Specific examples provided below, but are not intended to be an exhaustive list. In particular, attention should be given to the abstract and introduction to ensure that they set up the study to help the reader understand what was done and why it matters.

Response: Thank you for your suggestions. We have revised the text and figures to improve clarity, flow, and detail, ensuring they are more accessible to readers.

Abstract:

“A distinctive prediction of our theory is that spatial context in images modulates noise correlations, and the sign of the modulation depends on the overlap between receptive fields.” Suggest changing “spatial context” to “image size” – this is a more specific description of what was tested as there isn’t any breakdown or discussion of the results in terms of specific types of images and how they might vary in terms of the scale over which one part of the image is correlated with another.

Response: While we agree that “image size” is a specific and accurate descriptor of the manipulation tested in our study, we feel that “spatial context” better captures the broader conceptual framework of our theory. Spatial context refers to the structure and correlations within the visual input that influence neural variability and correlations. Image size is one specific manipulation that alters spatial context by changing the extent of the visual information surrounding the receptive fields. To address your comment and enhance clarity, we now include both terms, emphasizing the connection between spatial context and image size.

Manuscript changes: (Abstract)

Our analysis shows that spatial context in images reduces shared uncertainty for overlapping features, whereas it reduces independent uncertainty for non-overlapping features. As a result, the model predicts that increasing the size of an image reduces correlations for pairs with overlapping receptive fields and increases correlations for pairs with offset receptive fields. This prediction was confirmed by recordings from male macaque primary visual cortex.

Manuscript changes: (Discussion Line 475)

We have proposed a normative theory of V1 encoding of natural visual inputs, and empirically tested predictions for modulation of V1 covariability by spatial context (as manipulated by image size).

R3.4: “This is because spatial context reduces shared uncertainty (a source of shared variability) about overlapping image features, but it reduces private uncertainty about non-overlapping features.” I don’t understand the phrasing of this sentence. Specifically, “context reduces shared uncertainty. . . but it reduces private uncertainty”. While part of the problem might be that one of the “reduces” should be “increases”, I think the whole sentence is a bit unclear, because it’s too distant from what was actually done in the paper. Suggest replacing with something that specifically describes the use of shared vs independent modulator signals and the impact on the results.

Response: We agree that the original sentence was unclear and could be interpreted as overly abstract or distant from the main findings. We have replaced the sentence with a revised version that more directly describes the mechanisms in the shared and independent modulator models and their impact on the results.

The revised sentence now reads:

Manuscript changes: (Abstract)

According to the theory, variability reflects uncertainty about those latent features. In natural images, some sources of uncertainty are shared between features and lead to covariability between neurons, whereas other independent sources contribute to private variability. Our analysis shows that spatial context in images reduces shared uncertainty for overlapping features, whereas it reduces independent uncertainty for non-overlapping features.

This updated phrasing clarifies how spatial context impacts variability in the two models and aligns more closely with the work described in the paper.

R3.5: “Analysis of macaque V1 responses to natural images supports our theory, revealing that surround modulation switches from suppressing to enhancing noise correlations as the spatial offset between receptive fields grows.”. The terms “suppressing” and “enhancing” are not clear at this point of the paper because noise correlations can be either positive or negative. Would be better to stick close to the findings themselves – I assume this refers to figure 5, so something like “noise correlations are more positive for small images than large images in pairs of neurons with similar receptive field locations and orientation tuning preferences, but the noise correlations are more positive for larger vs smaller images when the two neurons have different receptive field locations and/or orientation tuning”. (That’s not a great suggestion; hopefully you can come up with a crisper way of putting it.)

Response: We agree that the terms “suppressing” and “enhancing” could benefit from additional clarity. To address this, we revised the sentence to describe the findings more explicitly while maintaining a concise and precise structure. The updated text now reads:

Manuscript changes: (Abstract)

As a result, the model predicts that increasing the size of an image reduces correlations for pairs with overlapping receptive fields and increases correlations for pairs with offset receptive fields. This prediction was confirmed by recordings from male macaque primary visual cortex.

R3.6: Introduction: I think the introduction would be more helpful to the reader if it arrived at the central idea of this study sooner. The shared vs. independent modulator signals don’t appear until we are about 3 paragraphs in to the Results section – I think this should come much earlier. To me, the key points the introduction should cover are: (1) we know that noise correlations matter to the important problem of inferring visual images from activity in V1; (2) with natural images, there are correlations in the images themselves across space that may be relevant to

V1's representation; (3) the brain might possibly be wired up to anticipate such correlations by having sources of modulation deployed across some spatial scale across the V1 representation; (4) here we test a simple version of this idea using a previously established generative model configured in two particular ways, with shared vs. independent sources of modulation, and we test the impact of these two configurations on (a) how well image identity can be inferred from the V1 activity patterns (b) how noise correlations vary as a function of neuron pair and image size.

I'm sure that summary isn't exactly right! But I hope it's helpful way of seeing what this reader took away from your study.

Response: Thank you for your comment. We have revised the Introduction to present the central idea of the study earlier, incorporating the key points you outlined to better frame the motivation and focus of our work. In particular, we added the following paragraph:

Manuscript changes: (Introduction Line 98)

We hypothesize that inferences about pairs of features should be modulated by spatial context in images, because contextual stimuli reveal new information about a stimulus and therefore reduce uncertainty. Importantly, our analysis of natural image statistics indicates multiple sources of uncertainty: Some sources are shared between features (Fig. 1B, top) and thus induce correlated variability, whereas others are independent (Fig. 1B, bottom) and thus induce independent variability. We further show that similar and overlapping features tend to have shared uncertainty, hence pairwise correlations between neurons encoding those features are reduced as the image is made larger by adding spatial context. Conversely, larger images reduce independent variability, thus increasing correlations, between neurons with less overlapping features. This prediction is strongly supported by our analysis of macaque V1 responses to natural images.

R3.7: Results: “For the pairwise application considered here, we reasoned that joint inferences about pairs of features should take into account whether those features are a priori more likely to be part of the same or different visual objects,” I like this insight a lot – can you say more? (perhaps not here, perhaps in the discussion?) How do you envision the brain knowing ahead of time whether two features are part of the same object and therefore deploying the same modulator?

Response: We have expanded the text to provide additional detail, addressing also a similar point raised by R1 (minor comments section). Regarding the mechanisms that could implement this prior grouping of neurons, we suggest one possibility in the Discussion section “Relation to stochastic divisive normalization and implications for circuit mechanisms of V1 covariability”. We refer to published circuit models that produce neural activity consistent with the inferences in our model (and prior publications) with the shared modulator. “It is plausible that image-computable versions of both frameworks will capture surround suppression of covariability that we observe here for overlapping RFs. The facilitation we observe for non-overlapping pairs may require additional tuning of the recurrent connectivity, to generate ensembles of neurons that effectively share normalization signals within each ensemble but not across ensembles.”

Manuscript changes: (Results Line 155)

The use of these two distinct GSMs was motivated by observations on the statistical properties of natural images, which often exhibit non-stationarities: namely, statistical dependencies can vary across regions. Due to these non-stationarities, features within the same visual object tend to be statistically dependent, influenced by common underlying factors. These dependencies are effectively captured by a GSM model with a single global modulator that scales all features of an object. In contrast, features belonging to different visual objects are generally more independent, as they are

influenced by separate factors Coen-Cagli et al. (2009). For such cases, using independent modulators that scale each feature separately, better captures the statistical independence of those features.

For the pairwise application considered here, we reasoned that joint inferences about pairs of features (represented by pairs of neurons) should account for whether these features are *a priori* more likely to be part of the same or different visual objects. This prior probability depends on factors such as similarity (e.g., orientation preference) and spatial proximity. Intuitively, features that are similar and located close together are more likely to belong to the same object and are thus better modeled with a shared global modulator. Conversely, dissimilar or distant features are more likely to belong to different objects, making independent modulators more suitable. By incorporating this reasoning into our model, we aim to capture the statistical dependencies (or lack thereof) between different features in natural images. Next, we tested this intuition formally.

R3.8: Figure 1: the shared/independent modulator signal is a very tiny part of the figure – can this be enlarged to indicate that this is the key focus of the paper? Also, please stay away from yellow – the noise inputs were not visible in the left part of 1B. Also, my understanding is that the noise inputs are independent for each neuron, but the yellow boxes in 1B (right side) draw them as if they are shared.

Response: We have enlarged the labels for “shared” and “independent” modulator signals in the figure to highlight their importance. Additionally, we changed the color of the input noise from yellow to brown to improve visibility.

The yellow boxes in Figure 1B were meant to represent the source of shared additive input noise, which contributes variability that is correlated between neurons (see **R3.2** above) and unrelated to the image content. In both models, the input noise remains unchanged between small and large images.

R3.9: Figure 2 and associated text: The purple/green panels in Fig 2B were difficult to follow because there appears to be some typo in the label for the color bar, with a “<” substituting for a “+”? Adding a common language description to the label would also help “Shared > Independent”, “Independent > Shared” for example.

Response: Thank you for pointing this out. We have corrected the typo in the color bar label of Fig. 2B and added a common language description (“shared > independent” and “independent > shared”) to clarify the color meanings. To address the issue of visualizing differences in negative values, we clipped the highest positive values to a threshold (>0.03), with all values above this threshold displayed in the same color. Additionally, we introduced a break in the color bar to make this adjustment more apparent. These changes should make the figure easier to interpret.

R3.10: As a general rule, when describing figures, it can be very handy to say directly to the reader what exactly in the figure supports the point. There’s an opportunity to do this in the text around line 200: “The 2D likelihood maps reveal that the shared GSM model largely outperforms the independent model when two neurons have overlapping RFs and similar orientations” could read: “The 2D likelihood maps reveal that the model with a shared modulator signal largely outperforms the model with an independent modulator signal when two neurons have overlapping RFs and similar orientations (green squares occupy a larger portion of the grid in Figure 2B left most panel vs a much smaller portion in the rightmost panel) ”.

Response: We have revised the text to directly guide the reader in interpreting the figure, making the connection between the description and the visual evidence in the figure clearer.

Manuscript changes: (Results Line 217)

The 2D likelihood maps reveal that the shared modulator model largely outperforms the independent modulator model when two neurons have overlapping receptive fields and similar orientations (as shown in Figure 2B, where green squares occupy a much larger portion of the grid on the left compared to the right).

R3.11: Line 228: “rsc” as an abbreviation for noise correlations is first used here, I think. While this term is standard in the field, it would be very confusing here for anyone not steeped in this topic because “spike count” as the binding for the “sc” part is not used. I suggest switching to “rnoise” throughout (since later you have “rsignal”).

Response: We acknowledge that this abbreviation may not be immediately intuitive for all readers. To address this, we have replaced ‘noise correlations’ with ‘correlations’ in the figures and text, and revised ‘signal correlations’ to ‘tuning similarity’ to prevent confusion with the term signal correlations. Additionally, we added an explanation in the manuscript clarifying that r_{sc} refers to spike count correlations, which is the Pearson correlation of spike count responses to repeated presentations of identical stimuli under the same behavioral conditions.

Manuscript changes (Line 241):

We next asked if the covariability between neuron pairs differs for small and large images, focusing on correlations (often referred to as spike count correlations, noise correlations, or r_{sc} , which measure the Pearson correlation of spike count responses across repeated identical stimuli) as is commonly done to measure changes in covariability beyond those due to changes in single-neuron variance.

R3.12: Line 242: “In the shared model, uncertainty about the global modulator is the *main* source of shared variability among neurons”. Isn’t it the *only* source of shared variability among neurons, at least for repeated presentations of the same stimulus? If this is not the case, then it should be clarified earlier (I note that if I’ve missed this then that would account for my confusion about Figure 3 noted above).

Response: Addressed in R3.2.

R3.13: “Centered” vs “off-centered” vs. “mixed” – are “off-centered” and “mixed” synonyms? Suggest picking one and sticking to it.

Response: The terms “mixed” (related to neural pairs) and “off-centered” (single neurons) are not synonymous. “Centered” refers to neurons covered by the small image, “off-centered” refers neurons covered by the large image but not the small image. “Mixed” refers to neuron pairs where one neuron is “centered” and the other is “off-centered”, as shown in the schematic in Fig. 4A.

R3.14: Page 5 – results in Figure 3 for simulations, results from V1 recordings – the magnitudes of the observed correlations are wildly different. This didn’t particularly worry me, but might confuse other readers, so it might be worth a mention somewhere that the size of the correlations in the simulations is (presumably) a function of the experimenter-chosen values for the modulator signal.

Response: In the original submission, we addressed the differences between the simulations and V1 recordings in the Discussion section (Line 551) in the revised manuscript). We have now added a pointer to that Discussion when we first report correlations values in the model (Line 253).

Manuscript changes: (Results)

Specifically, increasing stimulus size decreased correlations from 0.96 to 0.78 in the shared mode (Fig. 3B), but increased it from 0.08 to 0.28 in the independent model (Fig. 3D; see Discussion and Supplementary Fig. 2 for considerations about the magnitude of correlations in the simulations versus typical V1 data).

R3.15: Figure 4. “mixed pairs” used here, should this be “off-centered” pairs?

Response: Please refer to point **R3.13**. Additionally, “off-centered pairs” refers to neuron pairs where both neurons are off-centered, which is not the focus of this study.

I hope these comments are helpful. I think the work is interesting and valuable.

Response: Thank you for your positive evaluation and remarks.

References

- Aitchison, L. and Lengyel, M. (2016). The hamiltonian brain: Efficient probabilistic inference with excitatory-inhibitory neural circuit dynamics. *PLoS computational biology*, 12(12):e1005186.
- Bányai, M. and Orbán, G. (2019). Noise correlations and perceptual inference. *Current Opinion in Neurobiology*, 58:209–217.
- Bányai, M., Lazar, A., Klein, L., Klon-Lipok, J., Singer, W., and Orbán, G. (2019). Stimulus complexity shapes response correlations in primary visual cortex. *Proceedings of the National Academy of Sciences*, 116(7):2723–2732.
- Bondy, A. G., Haefner, R. M., and Cumming, B. G. (2018). Feedback determines the structure of correlated variability in primary visual cortex. *Nature Neuroscience*, 21(4):598–606.
- Cavanaugh, J. R., Bair, W., and Movshon, J. A. (2002). Selectivity and spatial distribution of signals from the receptive field surround in macaque v1 neurons. *Journal of neurophysiology*, 88(5):2547–2556.
- Coen-Cagli, R., Dayan, P., and Schwartz, O. (2009). Statistical models of linear and nonlinear contextual interactions in early visual processing. *Advances in neural information processing systems*, 22.
- Csikor, F., Meszema, B., and Orban, G. (2023). Top-down perceptual inference shaping the activity of early visual cortex. *bioRxiv*, pages 2023–11.
- Doulgeris, A. P. and Eltoft, T. (2009). Scale mixture of gaussian modelling of polarimetric sar data. *EURASIP Journal on Advances in Signal Processing*, 2009:1–13.
- Echeveste, R., Aitchison, L., Hennequin, G., and Lengyel, M. (2020). Cortical-like dynamics in recurrent circuits optimized for sampling-based probabilistic inference. *Nature neuroscience*, 23(9):1138–1149.
- Festa, D., Aschner, A., Davila, A., Kohn, A., and Coen-Cagli, R. (2021). Neuronal variability reflects probabilistic inference tuned to natural image statistics. *Nature communications*, 12(1):1–11.
- Goris, R. L., Coen-Cagli, R., Miller, K. D., Priebe, N. J., and Lengyel, M. (2024). Response sub-additivity and variability quenching in visual cortex. *Nature Reviews Neuroscience*, 25(4):237–252.
- Haefner, R. M., Berkes, P., and Fiser, J. (2016). Perceptual decision-making as probabilistic inference by neural sampling. *Neuron*, 90(3):649–660.
- Hashemi-Nezhad, M. and Lyon, D. C. (2012). Orientation tuning of the suppressive extraclassical surround depends on intrinsic organization of v1. *Cerebral Cortex*, 22(2):308–326.
- Lange, R. D. and Haefner, R. M. (2017). Characterizing and interpreting the influence of internal variables on sensory activity. *Current opinion in neurobiology*, 46:84–89.
- Orbán, G., Berkes, P., Fiser, J., and Lengyel, M. (2016). Neural variability and sampling-based probabilistic representations in the visual cortex. *Neuron*, 92(2):530–543.
- Roelfsema, P. R. (2023). Solving the binding problem: Assemblies form when neurons enhance their firing rate—they don’t need to oscillate or synchronize. *Neuron*, 111(7):1003–1019.
- Russakovsky, O., Deng, J., Su, H., Krause, J., Satheesh, S., Ma, S., Huang, Z., Karpathy, A., Khosla, A., Bernstein, M., et al. (2015). Imagenet large scale visual recognition challenge. *International journal of computer vision*, 115:211–252.
- Walker, G. A., Ohzawa, I., and Freeman, R. D. (1999). Asymmetric suppression outside the classical receptive field of the visual cortex. *Journal of Neuroscience*, 19(23):10536–10553.

Yatsenko, D., Josić, K., Ecker, A. S., Froudarakis, E., Cotton, R. J., and Tolias, A. S. (2015). Improved estimation and interpretation of correlations in neural circuits. *PLoS computational biology*, 11(3):e1004083.

Reviewer 1

Remarks to the Author:

The authors have thoroughly and exhaustively addressed all my comments and I believe that with the proposed update of the manuscript is ripe for publication. I especially appreciate the novel insight that supplementary figure 18 provides (produced in response to Reviewer 2). I would encourage the authors to present this under the results as it shows a true theoretically motivated property of models of V1. Also, I encourage the authors to include the analysis on diverse surround neurons, which is now shown in the response to reviewers, in the manuscript.

Response: We are grateful for your comments that thoroughly enhance the manuscript.

Regarding the Supplementary Figure 18, we agree with the reviewer that it provides important theoretical insight and better formalizes the intuitive schematics of Fig. 3E,F. As suggested, we now report this with new text early on in the Results.

Regarding the diverse surround neurons, we have incorporated the analysis in the manuscript as a new supplementary figure.

Supplementary Figure 21: Effect of surround filter orientation on correlations modulation. Changing the orientation of surround filters alters the level of correlations modulation for individual natural images (thin lines) due to their distinct structures. However, when the modulation is averaged across multiple natural images, the overall level of modulation remains similar (thick lines and circles). The orientations of the two neurons are specified as ‘orientation 1/orientation 2,’ where the first orientation corresponds to neuron 1 and the second to neuron 2. The filled gray circles correspond to the configuration used in Fig. 3G of the manuscript.

Reviewer 2

Remarks to the Author:

The revised manuscript is much improved, especially the exposition and figures. I had originally raised a number of concerns about methods, controls, and presentation, most of which were addressed. In the revised manuscript, some of my quantitative concerns remain. However, these concerns will almost certainly not affect the qualitative conclusions of the paper. These

quantitative concerns also could be levied against many comparable papers in the literature. So, I leave it to the editor to decide whether these concerns warrant further revisions.

Response: We appreciate your thoughtful comments and are glad that our revisions have improved the manuscript. As suggested, we have re-analyzed the data and enhanced the quantitative significance tests to address your remaining concerns. Additionally, to clarify why some of those concerns, while valid in general, do not apply to our analysis, we provide additional details of our recordings and analysis.

Lingering minor quantitative concerns

R2.1: I originally raised a minor concern having to do with EM vs moment-matching. I think the authors mistook this as a concern about having two separate models vs a single combined mixture model.

Allow me to restate the (minor) concern. The models are first fit to natural images using a moment-matching method to estimate the prior covariances. The shared-modulator and independent-modulator models are then compared (in Fig 2) by comparing their marginal likelihoods in various configurations. Let Σ_{MM} be the moment-matched prior covariance, and let Σ_{MLE} be the prior covariance one would get using MLE. I suspect (but might be wrong?) that Σ_{MLE} and Σ_{MM} are very similar matrices, but not exactly equivalent. My concern was then that $\log[p_{shared}(x|\Sigma_{MM})]$ is not exactly equivalent to $\log[p_{shared}(x|\Sigma_{MLE})]$, and likewise for $p_{independent}$. Again, I suspect that the effect on the conclusions of the paper will be nil even if the matrices are distinct.

If it's the case that Σ_{MM} is equivalent to Σ_{MLE} , I'd suggest a minor edit around L198 to clarify that the comparisons in Fig 2 are, in fact, comparisons of *maximum* marginal likelihoods.

Response: The reviewer raised a minor concern regarding the covariance matrices estimated via moment-matching (Σ_{MM}) versus those estimated using Maximum Likelihood Estimation (MLE) (Σ_{MLE}). To address this, we conducted additional analyses directly comparing the log-likelihood values obtained from both methods. Specifically, we estimated prior covariance matrices for the shared and independent GSM models using MLE, and compared the resulting log-likelihoods to those from the moment-matched covariances. The differences were minimal and did not affect the main patterns or conclusions. Thus, the marginal likelihood comparisons in Figure 2 remain robust, supporting the equivalence of the two estimation methods in capturing natural image statistics (new Supplementary Figure 20). As an additional control, we conducted a comparison with random covariance matrices and log-likelihood ratio between shared and independent GSM is nearly independent of the RF separation and orientation preferences (Reviewer Figure below), underscoring the importance of accurate covariance estimation.

We have now clarified this point explicitly in the manuscript (around line 198), highlighting that the likelihood comparisons indeed reflect maximum marginal likelihood conditions.

Manuscript changes: (Results)

The model parameters, i.e. the prior covariance matrices of the local latent features, were estimated with moment matching Dougeris and Eltoft (2009) from an ensemble of 10,000 natural images from the ImageNet validation set Russakovsky et al. (2015). This moment-matched covariance matrix is practically equivalent to the maximum likelihood estimate (MLE), and thus the resulting marginal likelihoods in Fig. 2 can be interpreted as approximating maximum marginal likelihoods.

Supplementary Figure 20: Estimating prior covariance matrices using Maximum Likelihood Estimation (MLE) yields log-likelihood patterns for natural images that are highly similar to those obtained with moment-matching. Slight discrepancies between MLE and moment-matching results (Supplementary Figure 7) may arise from differences in initialization and optimization criteria; however, these differences are minimal and do not affect the primary conclusions regarding model comparisons.

Estimating log-likelihoods with randomly generated covariance matrices produces a uniform distribution of log-likelihood ratios across different spatial locations and orientation preferences. This underscores the necessity of appropriately estimating covariance matrices from natural image statistics to accurately determine the optimal pairwise GSM structure.

R2.2: I originally raised a concern about "Insufficient details in the experimental methods and reporting of differences between subjects." I separately raised a concern that "I am wary of statistical tests on correlations..." Allow me to revisit both of these concerns together. The authors have partially addressed the first concern by including session details in Fig 4B and Table 1, and they have partially addressed the second concern by including Supplemental Fig 19.

Lingering concern R2.2a: Table 1 shows that subject a2 had 6 repeated sessions, while the other three subjects each sat for a single session. This raises a statistical concern in how the quantities in and around Fig 4 are calculated: repeated sessions from the same chronically-implanted Utah array do not constitute independent data points, but are "pseudo-replicated" data points. The t-tests that the authors use are only appropriate for independent data. Where the authors report values of $p < 0.001$, it is plausible to me that a corrected estimate might be $p < .05$ or even $p < .01$. The methods as stated suggest inflated significance due to pseudo-replication of a2's data. Again, I don't suspect the qualitative conclusions will change, but I do suspect that the *statistical significance* of the reported changes in correlations are inflated. A simple way to address this would be to report effect sizes and significance separately for each session. This has the reverse effect of under-inflating significance, but this is arguably preferable from a replicability standpoint. Another way to address it would be to choose any one of a2's sessions at random when calculating effect sizes for combined subjects.

Lingering concern R2.2b: analogous to the previous concern, doing t-tests on mean correlation values makes a similar error. In my first review, I fear I may have made some misleading comments about how to address this. I appreciate the authors' extensive efforts to fit regularized covariance estimates to their data (L1033-L1036). Unfortunately, this missed my point. The root of my concern remains in Fig 4a and in the p-values reported in L355 and L358. This is again a concern about inflated significance due to using a statistical test that assumes independence where the data are not actually independent. What the low-rank estimator of Yatsenko et al. provides is not just an alternative way to calculate correlations, but a way to approximate the number of degrees of freedom in the analysis. If the (mean-matched) *changes* to the noise covariance structure between small- and large- images is well-described by some rank- k change to the covariance, then the true degrees of freedom are more like $n * k$ for a population of n neurons. The current methods in the manuscript, as I understand them, use a t-test to compare mean correlations by taking all $n(n - 1)/2$ off-diagonal elements of the correlation matrix.

A quick-and-dirty correction would be to perform a t-test using $n * k - 1$ degrees of freedom rather than $n(n - 1)/2$, using $k \approx 3$ since that's what the authors report in their rebuttal as the lowest rank which gives good fits.

Another simple approach might be to design some sort of data-shuffling technique and use a permutation test, but the authors would need to think carefully about what shuffling method would be appropriate.

Finally, a third alternative would be to reframe the hypothesis entirely as a prediction for changes to the population *variance* along a particular dimension of response-space rather than a prediction for changes to the *mean correlation*. Indeed, "mean correlation" as a population statistic is only indirectly of theoretical interest. The directions and magnitudes of population noise-covariance are more directly relevant to signal detection theory. For this reason, I think that the analysis in Supp Fig 19 is both more statistically honest and more informative from a theory standpoint since it uses a *paired* test for changes in population *variance*.

I'm suggesting any of the above options. Option A is the authors might keep the paper as-is with some rough corrections to the reported p-values for mean correlations or, at the very least, a caveat in the text that the reported p-values are likely inflated. Option B is to do more thorough corrections through careful modeling of the null distribution for low-rank changes to covariance

or using a carefully constructed permutation procedure. Option C is to more substantially edit so that "mean correlations" are not the quantity of interest; essentially, replace the correlation analyses in Fig 4 and beyond with the population variance analyses in Fig 19 (further accounting for pseudo-replication across sessions, etc...). Option D is that the authors come up with their own sensible approach to appropriately "deflate" their statistical tests.

Response: We appreciate your detailed and thoughtful feedback. We have conducted new analyses along the lines of option A (details below) that, as predicted by the reviewer, confirm the original results despite the corrected p-values. Before explaining the details, here is a short summary: the p-values reported in the main text for aggregate data remain unchanged, as they are all well below the threshold of $p < 0.001$. Furthermore, when analyzing individual sessions, all but two of the mixed sessions meet the threshold of $p < 0.05$. The exact p-values and effect sizes are now reported in Table 1 in the revised manuscript.

First, we would like to explain why the reviewer's concern about pseudo-replicates (across sessions) does not apply to our datasets. It is true that there are 6 sessions from a single animal; however, each session involves different visual inputs and neurons. First, half of those sessions are obtained from an array implanted in the right hemisphere of V1, the other half from the left hemisphere, hence these are entirely different neurons. Second, in each session from the same hemisphere, we placed the stimuli in a different location of the visual field, such that approximately one fourth of the neurons had RFs centered on the stimuli and therefore were driven by the small stimuli (see Fig. 4A top-left for one example session). This means that 1) each neuron received different visual inputs across sessions even if the same image appeared on the screen; 2) in each session, a different subpopulation was driven by the small stimuli and therefore labeled as 'centered neurons'. In summary, it is correct to report results for across-session aggregate data in the main text. Nonetheless, we have now added the effect size and p-value separately for each session in Table 1, although this under-inflates significance as noted by the reviewer. In the caption of that table, we have also added the detailed explanation about the six sessions of a2.

Second, we would like to clarify why the 'inflation' of the original p-values is less dramatic than the reviewer might have predicted, even on a session-by-session basis. We estimate one correlation matrix per image, and we do so not for all N neurons on the array, but only for the $M_{\text{image}} < N$ neurons that are visually responsive to that particular image (inclusion criteria detailed in Methods). Therefore, when assessing p-values for mean correlation differences, in the original manuscript we used $M_{\text{image}}(M_{\text{image}} - 1)/2 \ll N(N - 1)/2$ pairs per image. On average across sessions and images, $N = 63$ whereas $M_{\text{image}} = 12$ hence the value of $M_{\text{image}}(M_{\text{image}} - 1)/2$ is typically not largely different from $(M_{\text{image}} \times K) - 1$.

With these clarifications in mind, now we provide the details of how we computed the corrected p-values for the revised manuscript. We randomly subsampled $M_{\text{image}} \times K$ pairs—where M_{image} is the number of responsive neurons per image and session, and K is based on the low-rank approximation from factor analysis—from the full set of $M_{\text{image}}(M_{\text{image}} - 1)/2$ possible neuron pairs. This procedure was repeated 1,000 times, and we report the average p-values across these repetitions.

Manuscript changes: (Methods section)

To avoid inflating significance due to the fact that the entries of an estimated noise covariance matrix are not independent, we proceeded as follows. In each session with N recorded neurons, for each image we computed the correlation matrix for the M_{image} neurons included for that particular image (see inclusion criteria above). We then randomly subsampled $M_{\text{image}} \times K$ pairs—where K is the estimated dimensionality of the covariance matrix based on the low-rank approximation from factor analysis (Yatsenko et al. (2015))—from the full set of $M_{\text{image}}(M_{\text{image}} - 1)/2$ possible neuron pairs. Lastly, we aggregated the sampled pairs across images and sessions to compute a p-value. This procedure was repeated 1,000 times, and we report the average p-values across these repetitions. (See also Table 1 for session-by-session significance).

Lastly, as explained in the first response letter, we agree that the population-level analysis of option C is interesting

on its own, and we will pursue it further in follow-up work. Here we have decided to focus on mean correlations also for comparability with numerous previous studies that are similarly focused.

Table 1: Details of the nine recording sessions across four animals. Note that there are 6 sessions from a single animal, a2; however, each session involves different visual inputs and neurons. First, half of those sessions are obtained from an array implanted in the right hemisphere of V1, the other half from the left hemisphere, hence these are entirely different neurons. Second, in each session from the same hemisphere, we placed the stimuli in a different location of the visual field, such that approximately one fourth of the neurons had RFs centered on the stimuli and therefore were driven by the small stimuli (see Fig. 4A top-left for one example session). This means that 1) each neuron received different visual inputs across sessions even if the same image appeared on the screen; 2) in each session, a different subpopulation was driven by the small stimuli and therefore labeled as “centered neurons”.

animal	session	recording type	cent. neuron	off-cent. neuron	cortical layer	Cohen’s d (cent.)	Cohen’s d (mix.)	p-val (cent.)	p-val (mix.)
a1	s1	Utah Array	25	32	L2/3 or 4B	0.16	-0.03	3.7×10^{-18}	0.3
a2	s1	Utah Array	45	52	L2/3 or 4B	0.12	-0.02	7.5×10^{-19}	0.2
a2	s2	Utah Array	39	53	L2/3 or 4B	0.11	-0.08	9.8×10^{-10}	3.8×10^{-3}
a2	s3	Utah Array	40	26	L2/3 or 4B	0.1	-0.25	1.6×10^{-12}	3.4×10^{-8}
a2	s4	Utah Array	49	41	L2/3 or 4B	0.13	-0.05	4.3×10^{-19}	2×10^{-2}
a2	s5	Utah Array	22	52	L2/3 or 4B	0.28	-0.94	3.4×10^{-49}	0
a2	s6	Utah Array	50	49	L2/3 or 4B	0.23	-0.16	8×10^{-59}	2.6×10^{-15}
a3	s1	Utah Array	42	61	L2/3 or 4B	0.05	-0.3	1×10^{-2}	6.2×10^{-19}
a4	s1	Neuropixels	53	190	All layers	0.04	-0.05	4×10^{-2}	3×10^{-3}

Concerns that were addressed

Replicating (sanity-checking) some key marginal statistics of the model: now done in Supp Fig 16. (It would be useful to state that the ‘single neuron’ analyzed is representative of the population).

Validating that models seen in prior work *do not* show the key effects: now done in Supp Figs 14-15. This is done using ablations on the present model, which I think is a nice way to address the original concern while highlighting the innovation here.

Missing references are now included, and differences with the present work are nicely explained.

My original comment about MAP vs full-inference over modulators is addressed. I was mostly wrong about Simpson’s paradox being the key to understanding the positive correlations. But I’m glad that this comment did lead to clearer interpretation of the source(s) of the observed effects in the Discussion and in Fig 18.

Figures are overall improved. I appreciate the use of contour plots to show the true posteriors rather than gaussian approximations.

Response: We appreciate your constructive feedback and are glad that the revisions addressed your concerns.

For the ‘single neuron’ analysis, we added the following sentence to the caption of Supplementary Figure 16: *The analyzed single neuron is representative of the neuronal population.*

Reviewer 3

Remarks to the Author:

I appreciate the hard work the authors have put in to revising this paper. It is much improved, but it is still quite hard to follow in places. My comments below include one key point and a series of suggestions for greater precision/clarity in phrasing/definitions.

Response: We are grateful for the positive evaluation. Please see our response to the comments below.

R3.1: The overarching finding of this paper is that the shared vs independent modulators operating on a V1-like circuit can produce different outcomes depending on the spatial scale of the images that the circuit is responding to. This raises the issue of how the brain ‘decides’ over what spatial scale (in V1) shared vs independent modulators might operate, since it can’t know a priori what the visual stimuli are. I raised this point in the previous review (R3.7 “How do you envision the brain knowing ahead of time whether two features are part of the same object and therefore deploying the same modulator?”). The authors have added a section to the Discussion regarding divisive normalization and normalization pools, but I don’t think this addresses the comment. Conceptually, I’m interested in the authors’ thoughts/speculations about whether they envision the shared/independent modulation scheme is adjusted dynamically based on some initial response to the visual stimuli, or is it hard wired at some scale across V1 representation based on how the circuit has developed in response to typical scene statistics, or something else? Ultimately, my comment is about making sure any critiques about potentially circular reasoning are considered and addressed.

Response: Thank you for raising this point. We would like to clarify that there is no circular reasoning in our proposal, because of the distinction between the Bayesian prior and posterior distributions in our computational model, and the corresponding, distinct mechanistic implementations that we point out in Discussion. First, we propose that local circuit architecture—e.g., closely spaced RFs—serves as a built-in Bayesian prior indicating that nearby features share a modulator, whereas distant features rely on independent modulators, on a spatial scale learned from exposure to typical image statistics. This does not require the system to know a priori which features belong to the same object in one specific visual input; rather, it reflects a configuration that is generally consistent with natural image statistics, and therefore is expected to be valid for the majority of images. As discussed in lines 584–604, circuit models of normalization suggest how a fixed, hard-wired distinction between shared and independent modulators might be implemented mechanistically.

Second, as we note in lines 540–550, these modulator assignments could also be reconfigured flexibly for each new visual input (although testing the effects of this reconfiguration in our data will require a follow-up study). In that scenario, the default hard-wired arrangement could be refined by feedback processes that dynamically adapt modulation based on the actual image context. In other words, the system begins with a “prior” assumption that works for most scenes but can be reshaped when needed by top-down or lateral feedback. We have added these considerations in the Discussion, L619.

R3.2: Outsiders to the field could still have difficulty understanding the reasoning behind the claims. I’ll try to be as specific as I can.

- a. Does the term “image” always refer to the visual stimulus, or does it (sometimes?) refer to the postulated inference about what the visual stimulus is based on the neural activity in V1?
- b. Does the term “modulator” refer to a postulated neural circuit element, a mathematical construct capturing correlated spatial context intrinsic to natural visual stimuli (i.e. part of the external world, not part of the postulated neural circuitry), or something else?
- c. Does the term “feature” refer to aspects of visual stimuli, aspects of V1 receptive fields, or?
- d. Introduction, lines 75-78: “here we consider a simple generative model known as Gaussian Scale Mixture (GSM29). The key assumption of this model is that a global ‘modulator’ variable modulates multiple features and thus introduces statistical dependence among them (details in Fig. 1A and in Results)”. What do ‘modulator’ and ‘features’ refer to here? Is the modulator a brain signal? Are the features attributes of the image, attributes of neural tuning functions, activity levels of neurons (maybe in multiple neurons), or something else entirely?

Response: In the revised manuscript we have made sure that we use ‘image’ always to refer to the visual stimulus, and that we use ‘feature’ (and equivalently ‘image feature’), and ‘modulator’ consistently, as explained next.

The GSM model assumes that an ‘image’ can be described as a weighted sum of ‘features’ (each feature is like an elementary image: a wavelet with a specific orientation and spatial frequency); that weighted sum is then multiplied by a positive scalar variable (the ‘modulator’), as illustrated in Figure 1A. This is a mathematical description of how an image may be generated, and those variables are mathematical constructs. Each choice of specific values for the coefficients of the weighted sum and for the modulator, will generate a specific image.

The problem faced by V1 neurons, we hypothesize, is the inverse problem: when observing an image (the visual input) we assume that V1 neurons infer the values of the coefficients that are most likely to have generated that particular image. The postulated inferences are about the coefficients of that weighted sum, which we term ‘latent variables g ’. Equivalently, we sometimes write ‘inference about image features’ because the inference about a coefficient ‘ g ’ tells us how much the corresponding feature contributes to the image.

Just like the coefficients ‘ g ’, also the ‘modulator’ refers to a latent variable in the GSM. In the reviewer terminology, it is a mathematical construct, not a circuit element. Performing the inference about ‘ g ’ does not require to explicitly infer also the value of the ‘modulator’, instead it requires marginalizing it out (that is, solving an integral). In this manuscript we do not test, nor exclude, whether there is a circuit element (e.g. a neuron subtype or a specific circuit motif) tasked with explicitly representing inferences about the modulator.

Manuscript changes: (Results and Methods sections)

A summary of the generative process of the Gaussian scale mixture (GSM) model. The linear transform of the raw image pixel values, denoted as \mathbf{x} , results from combining local oriented features (in pink and cyan), each weighted by a Gaussian coefficient \mathbf{g} .

The GSM model for a single neuron (Fig. 1A) assumes that an image is generated from linear combinations of localized oriented features (each feature is like an elementary image: a wavelet with a specific orientation and spatial frequency), each weighted by a Gaussian coefficient (a latent variable \mathbf{g}). A global modulator variable (latent variable v) scales multiplicatively that weighted sum, and noise (η) is added, resulting in the observed variable \mathbf{x} (related to the image by a linear transformation; see Methods). This is a mathematical description of how an image may be generated: each choice of specific values for the coefficients, the modulator, and the noise, will generate one specific image.

The problem faced by V1 neurons, we hypothesize, is the inverse problem: when observing an image (the visual input) we assume that V1 neurons infer the values of the coefficients that are likely to have generated that particular image. More precisely, we hypothesize that a V1 neuron encodes the posterior distribution of the Gaussian coefficient (\$g\$ ) associated with a target feature (Eqs. 1 and 2 in Methods).

Given our focus on pairwise V1 activity, our primary objective is to estimate the joint posterior probability distribution of the features encoded by two neurons, given an image, denoted as \$p(g_{c1}, g_{c2} | x)\$.

We used Bayesian inference to estimate the posterior distribution of latent variables of the pairwise GSM. Specifically, because we assumed that neural activity represents samples from the posterior distribution over local features \mathbf{g} , our objective was to compute the posterior $p(\mathbf{g} | \text{stimulus})$, which involves marginalization over v (Note that we do not test, nor exclude, whether there is a circuit element, e.g. a neuron subtype or a specific circuit motif, tasked with explicitly representing inferences about \$v\$.

e. (Minor) “Past work strongly supports this theory for single neuron activity” – does “this theory” refer to the Gaussian Scale Mixture model or the sampling hypotheses? Both are mentioned in the preceding paragraph.

Response: We clarified that “this theory” refers to the combination of the Gaussian Scale Mixture (GSM) model and the sampling hypothesis, which together form the theoretical framework underlying our analysis (L87): Past work strongly supports this theoretical framework—which combines a Gaussian Scale Mixture model of natural image statistics with the sampling hypothesis of neural representation—for explaining single neuron activity...

f. Results, first two paragraphs: “To study the relationship between pairwise neural responses and image statistics”...” In the single neuron GSM (Fig. 1A), an image is generated from linear combinations of localized oriented features, each modulated by a Gaussian coefficient (a latent variable g).”.

Does “image” here refer to a visual stimulus, or to some neural representation of that stimulus? The juxtaposition of ‘neural/neuron’ with ‘image’ in these two sentences leaves the meaning unclear.

Without understanding which we are referring to, it becomes impossible to be sure what is meant by ‘localized oriented features’, or what is being postulated as modulating those features. Are the ‘localized oriented features’ receptive fields of input neurons? Or does this text solely refer to visual stimuli, and how they were generated for testing of a neural model?

Figure 1’s legend didn’t provide clarification on these questions, but the Methods did provide some help, and some of these details should be incorporated.

Response: Please see response to the previous point for clarification of ‘image’ and ‘features’. In addition, we have modified the text quoted by the reviewer to avoid confusion with the ‘modulator’. We now write “In the **GSM** model for a single neuron” (Fig. 1A), an image is generated from linear combinations of localized oriented features, each **weighted** by a Gaussian coefficient (a latent variable g).”

These are some examples from early in the paper. If the reader is set up well for understanding early on, it will be worth the effort.

Response: Thank you for your comments.

References

References

- Doulgeris, A. P. and Eltoft, T. (2009). Scale mixture of gaussian modelling of polarimetric sar data. *EURASIP Journal on Advances in Signal Processing*, 2009:1–13.
- Russakovsky, O., Deng, J., Su, H., Krause, J., Satheesh, S., Ma, S., Huang, Z., Karpathy, A., Khosla, A., Bernstein, M., et al. (2015). Imagenet large scale visual recognition challenge. *International journal of computer vision*, 115:211–252.
- Yatsenko, D., Josić, K., Ecker, A. S., Froudarakis, E., Cotton, R. J., and Tolias, A. S. (2015). Improved estimation and interpretation of correlations in neural circuits. *PLoS computational biology*, 11(3):e1004083.